# Variance Reduction of Stochastic Hypergradient Estimation by Mixed Fixed-Point Iteration

**Naoyuki Terashita**     *naoyuki.terashita.sk@hitachi.com*
*Hitachi, Ltd.*
*Osaka University*

**Satoshi Hara**     *satohara@uec.ac.jp*
*University of Electro-Communications*

**Reviewed on OpenReview:** *https://openreview.net/forum?id=mkmX2ICi5c*

## Abstract

Hypergradient represents how the hyperparameter of an optimization problem (or inner-problem) changes an outer-cost through the optimized inner-parameter, and it takes a crucial role in hyperparameter optimization, meta learning, and data influence estimation. This paper studies hypergradient computation involving a stochastic inner-problem, a typical machine learning setting where the empirical loss is estimated by minibatches. Stochastic hypergradient estimation requires estimating products of Jacobian matrices of the inner iteration. Current methods struggle with large estimation variance because they depend on a specific sequence of Jacobian samples to estimate this product. This paper overcomes this problem by *mixing* two different stochastic hypergradient estimation methods that use distinct sequences of Jacobian samples. Furthermore, we show that the proposed method enables almost sure convergence to the true hypergradient through the stochastic Krasnosel'skiĭ-Mann iteration. Theoretical analysis demonstrates that, compared to existing approaches, our method achieves lower asymptotic variance bounds while maintaining comparable computational complexity. Empirical evaluations on synthetic and real-world tasks verify our theoretical results and superior variance reduction over existing methods. The code is available at `https://github.com/hitachi-rd-cv/mixed-fp`.

## 1 Introduction

Hypergradient is essential in a wide range of tasks, including bilevel optimization which encompasses hyperparameter optimization (Maclaurin et al., 2015; Franceschi et al., 2017; Liu et al., 2018; Lorraine et al., 2019) and meta learning (Andrychowicz et al., 2016; Finn et al., 2017; Franceschi et al., 2018), as well as an explainability technique called data influence estimation (Koh & Liang, 2017; Hara et al., 2019; Pruthi et al., 2020; Xue et al., 2021). Using a hyperparameter (or outer-parameter) $\lambda$ and an inner-parameter $x$ that parameterize a scalar cost $f$ and a vector mapping $\varphi$, the hypergradient can be defined as

$$\underbrace{\mathrm{d}_\lambda f(x(\lambda), \lambda)}_{\text{hypergradient}} \quad \text{with} \quad x(\lambda) = \varphi(x(\lambda), \lambda),$$

where $\mathrm{d}_\lambda f(x(\lambda), \lambda)$ represents the derivative of $\lambda \mapsto f(x(\lambda), \lambda)$, which takes the differentiation of $\lambda \mapsto x(\lambda)$ into account. A common choice for $\varphi$ is to set $\varphi(x, \lambda) = x - \gamma \partial_x g(x, \lambda)$ with $\gamma \in \mathbb{R}_{++}$ and a strongly-convex function $g$. In this case, $x(\lambda) = \varphi(x(\lambda), \lambda)$ is equivalent to the minimization problem $x(\lambda) = \arg\min_x g(x, \lambda)$.

Calculating exact hypergradients is prohibitive for large-scale problems as it requires the inverse of a parameter-sized matrix, making their efficient approximations crucial. A method called approximate implicit differentiation (AID) (Grazzi et al., 2020; Pedregosa, 2016; Lorraine et al., 2019; Rajeswaran et al.,

2019) considers the following form derived from the chain rule and implicit function theorem:

$$d_\lambda f(x(\lambda), \lambda) = \partial_\lambda \varphi (I - \partial_x \varphi)^{-1} \partial_x f + \partial_\lambda f,$$

where the arguments $(x(\lambda), \lambda)$ are omitted and $\partial_x f$ represents $\partial f / \partial x$. AID then replaces $x(\lambda)$ with the inexact solution and approximately solves the resulting linear system. A common AID approach called fixed-point method (Lorraine et al., 2019) uses a truncated Neumann series approximation with $M$ terms to solve the linear system,

$$(I - \partial_x \varphi)^{-1} \approx \sum_{m=0}^{M-1} \partial_x \varphi^m,$$

to avoid explicit calculation of the inverse matrix and Jacobian[1].

When the Jacobian matrix $\partial_x \varphi$ is stochastically estimated, a typical scenario where the empirical loss is estimated by minibatches, computing hypergradient becomes even more challenging. To perform such an estimation, Koh & Liang (2017) and Grazzi et al. (2021) used the stochastic fixed-point method which performs the following iteration with independent samples of $\partial_x \varphi$ denoted by $(\hat{A}_m)_{m \in \mathbb{N}}$:

$$\hat{w}_{m+1} = \hat{A}_m \hat{w}_m + \partial_x f. \tag{1}$$

After $M$ iterations, $\hat{w}_M$ estimates the truncated Neumann series as:

$$\hat{w}_M = \sum_{m=0}^{M-1} \left( \prod_{k=m}^{M-1} \hat{A}_k \right) \partial_x f \approx \sum_{m=0}^{M-1} \partial_x \varphi^m \partial_x f.$$

This method can be seen as estimating the Jacobian product by,

$$\partial_x \varphi^{M-m} \approx \hat{A}_{M-1} \hat{A}_{M-2} \cdots \hat{A}_{m+1} \hat{A}_m,$$

for each $m < M$. This approach incurs large estimation variance because each order of the Jacobian product is sampled only once using a specific sequence of Jacobian samples.

This work aims at reducing estimation variance by employing multiple different sequences of Jacobian samples to estimate any order of Jacobian product, without sacrificing computational complexity. We first point out that there are two hypergradient estimators that use different sequences of Jacobian samples to estimate their product: Stochastic fixed-point method (StocFP) (Koh & Liang, 2017; Grazzi et al., 2021) and Stochastic recurrent backpropagation (StocRB) (Ji et al., 2021; Yang et al., 2021). Next, we propose MixedFP, which updates a weighted average of StocFP and StocRB at each step. MixedFP realizes variance reduction by estimating the Jacobian product of a given order using various sample sequences. Furthermore, we improve our MixedFP so it converges almost surely to the true hypergradient by applying the stochastic Krasnosel'skiǐ-Mann iteration (Grazzi et al., 2021; Bravo & Cominetti, 2024).

Additionally, this paper quantifies the variance reduction performance of the proposed method both theoretically and empirically. Our analysis showed that the proposed method improves the expected error compared to the existing stochastic bilevel optimization methods (Grazzi et al., 2021; Ji et al., 2021; Arbel & Mairal, 2022) in terms of one-shot hypergradient estimation. Numerical experiments using real-world data also support the theoretical observations and demonstrate that proposed methods can estimate hypergradients more accurately than the existing methods.

The main contributions of this work are as follows:

- We point out that there are two algorithms in stochastic hypergradient estimation methods that estimate Jacobian products using different sequences of samples: Stochastic fixed-point method (StocFP) and Stochastic recurrent backpropagation (StocRB).

---

[1]For simplicity of exposition, here we assume that the exact inner solution $x(\lambda)$ is given. In practice, AID also approximates this solution.

- We propose MixedFP, which reduces the estimation variance by iteratively updating a weighted average of StocFP and StocRB.

- We achieve almost sure convergence to the true hypergradient by applying the stochastic Krasnosel'skiĭ-Mann iteration to MixedFP.

- Through theoretical analysis of error bounds, we demonstrate improvement over previous research (Grazzi et al., 2021; Ji et al., 2021; Arbel & Mairal, 2022).

- Our experiments using real-world data demonstrate a smaller estimation variance of the proposed methods compared to the existing approaches.

## 2 Related Work

Stochastic hypergradient estimation has been studied in the context of stochastic bilevel optimization (Ghadimi & Wang, 2018; Couellan & Wang, 2016; Ji et al., 2021) to minimize estimation variances of solutions for both inner- and outer-problems. Ji et al. (2021) and Yang et al. (2021) propose methods that focus on two-time scale updates and stepsize adjustments to accelerate inner-problem optimization with a warm-up strategy. However, these studies do not address the variance reduction of stochastic hypergradient estimation because the noise on the hypergradient estimation is manageable by decreasing stepsizes of their outer-optimization. In contrast, our work specifically aims at reducing the variance of hypergradient estimation itself. This is particularly important for tasks like influence estimation (Koh & Liang, 2017; Khanna et al., 2019), which estimates the impact of excluding training data on performance by a single-shot hypergradient estimation.

There are a few studies that addressed the variance reduction of hypergradient estimation. Grazzi et al. (2021) applied the stochastic Krasnosel'skiĭ-Mann iteration to (1), which is equivalent to solving $\min_v \frac{1}{2}\|(I - \partial_x \varphi)v - \partial_x f\|^2$ by the stochastic gradient descent (Arbel & Mairal, 2022), to achieve almost sure convergence to the true hypergradient. Moreover, Arbel & Mairal (2022) applies the warm-up to the hypergradient estimation to improve convergence properties. Our MixedFP incorporates their solution as a special case, enabling a more general and effective framework for stochastic hypergradient estimation. Note that this paper does not restrict the context of hypergradients to bilevel optimization, and therefore the warm-up strategy is not employed. Introducing a warm-up is a promising direction for future research, which can enhance the convergence of the bilevel optimization solved by our MixedFP.

## 3 Hypergradient and Stochastic Approximate Implicit Differentiation

In this section, we first redefine the hypergradient and then introduce two existing methods for computing stochastic hypergradients. We also highlight the difference in the sequences of Jacobian samples used for estimation in these methods.

### 3.1 Hypergradient and Its Approximation

The hypergradient $d_\lambda f(x(\lambda), \lambda)$ is the total derivative of the composite function $\lambda \in \mathbb{R}^{d_\lambda} \mapsto f(x(\lambda), \lambda) \in \mathbb{R}$, where $f(x, \lambda) \in \mathbb{R}$ is the outer-cost function and $x(\lambda) \in \mathbb{R}^{d_x}$ is the inner-parameter solution. The solution of the inner-parameter $x(\lambda)$ is defined as the fixed point of a mapping $\varphi : \mathbb{R}^{d_x} \times \mathbb{R}^{d_\lambda} \to \mathbb{R}^{d_x}$, which gives the hypergradient as

$$d_\lambda f(x(\lambda), \lambda) \quad \text{with} \quad x(\lambda) = \varphi(x(\lambda), \lambda).$$

Approximate implicit differentiation (AID) (Lorraine et al., 2019; Pedregosa, 2016) is a set of approaches consisting of the derivation of a linear system for $d_\lambda f(x(\lambda), \lambda)$ using the implicit function theorem along with two approximations: an approximate inner solution $x(\lambda)$ and an approximate linear system solution. The implicit function theorem and chain rule rewrite the hypergradient in a form that involves an inverse

matrix:

$$d_\lambda f(x(\lambda), \lambda) = \partial_\lambda \varphi \cdot d_x f + \partial_\lambda f, \tag{2a}$$

$$\text{where} \quad d_x f = (I - \partial_x \varphi)^{-1} \partial_x f. \tag{2b}$$

Here and hereafter, the arguments $(x(\lambda), \lambda)$ are omitted when clear from the context. Moreover, for any vector function $h : \mathbb{R}^m \to \mathbb{R}^n$, we denote its partial derivative by $\partial_x h(x) \in \mathbb{R}^{m \times n}$; e.g., $\partial_\lambda f$ represents the partial derivative of $f(x(\lambda), \lambda)$ with respect to the second argument, treating $x(\lambda)$ as a constant.

(2) is justified under the following assumption.

**Assumption 1.** *For every $\lambda \in \mathbb{R}^{d_\lambda}$, we assume:*

(i) $\varphi(\cdot, \lambda)$ *is a contraction; i.e., there exists a constant $\rho < 1$ such that $\|\partial_x \varphi(x, \lambda)\| \le \rho$ for any $x \in \mathbb{R}^{d_x}$.*

(ii) $\varphi(x, \lambda)$ *and $f(x, \lambda)$ are differentiable at any $x \in \mathbb{R}^{d_x}$.*

Since calculating the inverse matrix is expensive for a large $d_x$, Lorraine et al. (2019) proposed the fixed-point method, which approximates (2b) using a truncated Neumann series:

$$d_x f \approx \sum_{m=0}^{M-1} \partial_x \varphi^m \partial_x f. \tag{3}$$

This approximation becomes exact as $M$ approaches infinity. Hereafter, we consider a scenario where the exact inner solution $x(\lambda)$ is given, since our analysis in Appendix C indicates that the error from the inexact inner solution remains consistent among ours and previous work.

### 3.2 Stochastic Approximate Implicit Differentiation

From here on, we assume that we only have access to a stochastic estimator of $\varphi(x, \lambda)$ denoted by $\hat{\varphi}(x, \lambda; \xi)$, where $\xi$ is some random variable whose values lie within a measurable space $\mathcal{X}$. For convenience, we introduce the following notations:

$$A = \partial_x \varphi(x(\lambda), \lambda), \qquad \hat{A} = \partial_x \hat{\varphi}(x(\lambda), \lambda; \xi), \qquad \hat{A}_m = \partial_x \hat{\varphi}(x(\lambda), \lambda; \xi_m), \qquad c = \partial_x f(x(\lambda), \lambda),$$

where $(\xi_m)_{m \in \mathbb{N}}$ are independent copies of $\xi$. We assume that $\hat{\varphi}$ satisfies the following conditions:

**Assumption 2.** *For every $x \in \mathbb{R}^{d_x}$ and $\lambda \in \mathbb{R}^{d_\lambda}$,*

(i) $\hat{\varphi}(x, \lambda; \xi)$ *is an unbiased estimator of $\varphi(x, \lambda)$; i.e., $\mathbb{E}[\hat{\varphi}(x, \lambda; \xi)] = \varphi(x, \lambda), \forall x, \lambda$.*

(ii) $\hat{\varphi}$ *is differentiable with respect to the first and second arguments at any $\xi \in \mathcal{X}$.*

Assumption 2(i) can be easily satisfied in a typical machine learning scenario where the stochastic mapping is a stochastic gradient descent, i.e., $\hat{\varphi}(x, \lambda; \xi) = x - \gamma \partial_x \hat{g}(x, \lambda; \xi)$, and the instance $\xi$ is uniformly sampled from a finite dataset.

Note that, unlike common settings of stochastic bilevel optimization (e.g., Ghadimi & Wang (2018)), we do not consider estimation of $f$. This is solely for clarity, as our primary focus is on the estimation error caused by the inverse matrix approximation with stochastic $\hat{\varphi}$, and our primary distinction from the previous studies (Arbel & Mairal, 2022; Grazzi et al., 2020) is the improvement on this approximation.

#### 3.2.1 Stochastic Fixed-Point Method (StocFP)

The studies by Grazzi et al. (2021) and Koh & Liang (2017) employ an iteration that we call Stochastic fixed-point method (StocFP), a stochastic variant of the fixed-point method (Lorraine et al., 2019).

$$\hat{w}_0 = c \tag{4a}$$

$$\text{For } m = 0, \dots, M - 1:$$

$$\lfloor \hat{w}_{m+1} = \hat{A}_m \hat{w}_m + c \tag{4b}$$

The iteration (4b) accumulates the products of Jacobian samples and their sum simultaneously. Assumption 2 guarantees that $\hat{w}_M$ is an unbiased estimator of the right-hand side of (3). A key advantage of this method is that the vector-Jacobian product $\hat{A}_m w_m \in \mathbb{R}^{d_x}$ can be calculated in $O(d_x)$ time using reverse mode automatic differentiation (Baydin et al., 2018), because $\hat{A}_m w_m$ is the derivative of the scalar function $x \mapsto \varphi(x, \lambda; \xi_m)^\top w_m$ evaluated at $x(\lambda)$. The total computation time is therefore $O(Md_x)$ with a memory requirement of $O(d_x)$.

### 3.2.2 Stochastic Recurrent Backpropagation (StocRB)

Ji et al. (2021) independently proposed a method for estimating $d_x f$ using a different form of iterations:

$$\hat{y}_0 = 0 \in \mathbb{R}^{d_x}, \ \hat{u}_0 = c \tag{5a}$$

For $m = 0, \ldots, M - 1$:

$$\left|\begin{aligned} \hat{y}_{m+1} &= \hat{y}_m + \hat{u}_m \\ \hat{u}_{m+1} &= \hat{A}_m \hat{u}_m \end{aligned}\right. \tag{5b}$$

The final output $\hat{y}_M$ is an estimator of $d_x f$. Unlike StocFP (4b), this method divides the computation of (3) into the product of Jacobians estimated by $\hat{u}_m$ and their sum estimated by $\hat{y}_m$. We refer to (5) as Stochastic recurrent backpropagation (StocRB) because this method can be understood as backpropagation from $f$ with respect to $x$ that has recurrently passed through $\hat{\varphi}$ for $M$ times.

While both StocFP (4) and StocRB (5) provide the same hypergradient approximation in expectation, they yield different estimates given the same samples. This is because they estimate the expected Jacobian product using different sequences of Jacobian samples:

**Remark 1.** *For any $m = 1, \ldots, M$, StocFP estimates $A^m$ using $\hat{A}_{M-1} \cdots \hat{A}_{M-m}$, while StocRB uses $\hat{A}_{m-1} \cdots \hat{A}_0$ to estimate the same value:*

| Stochastic Fixed-Point Method (**StocFP**) (4) | Stochastic Recurrent Backprop. (**StocRB**) (5) |
|---|---|
| $\hat{w}_M = \quad c \qquad\qquad\qquad (= A^0 c)$ $\qquad + \hat{A}_{M-1} c \qquad\qquad (\approx A^1 c)$ $\qquad + \hat{A}_{M-1} \hat{A}_{M-2} c \qquad (\approx A^2 c)$ $\qquad \vdots$ $\qquad + \hat{A}_{M-1} \hat{A}_{M-2} \cdots \hat{A}_1 c \quad (\approx A^{M-1} c)$ $\qquad + \hat{A}_{M-1} \hat{A}_{M-2} \cdots \hat{A}_1 \hat{A}_0 c \quad (\approx A^M c)$ | $\hat{y}_M = \qquad\qquad\qquad c \quad (= A^0 c)$ $\qquad + \qquad\qquad\qquad \hat{A}_0 c \quad (\approx A^1 c)$ $\qquad + \qquad\qquad\qquad \hat{A}_1 \hat{A}_0 c \quad (\approx A^2 c)$ $\qquad \vdots$ $\qquad + \qquad \hat{A}_{M-2} \cdots \hat{A}_1 \hat{A}_0 c \quad (\approx A^{M-1})$ $\qquad + \hat{A}_{M-1} \hat{A}_{M-2} \cdots \hat{A}_1 \hat{A}_0 c \quad (\approx A^M c)$ |

## 4 Variance Reduction of Stochastic Approximate Implicit Differentiation

The proposed method is based on two ideas. First, in Section 4.1, we show that the estimation variance can be reduced by a weighted average-like iteration between StocRB and StocFP. Second, Section 4.2 explains that almost sure convergence to the true hypergradient is achieved by using the stochastic Krasnosel'skiĭ-Mann iteration (Bravo & Cominetti, 2024). For clarity, proofs of all theorems and lemmas are deferred in the appendix.

### 4.1 Mixed Fixed-Point Iteration (MixedFP)

Our MixedFP mixes StocFP and StocRB, motivated by the observation in Remark 1 that the sequences of Jacobian samples used for estimation are different from each other.

We introduce a parameter called the mixing rate $\alpha \in [0,1]$ and derive the following MixedFP algorithm:

$$\hat{v}_0 = 0 \in \mathbb{R}^{d_x}, \ \hat{w}_0 = \hat{u}_0 = c \tag{6a}$$

For $m = 0, \ldots, M-1$:

$$\left| \begin{array}{l} \hat{v}_{m+1} = \alpha(\hat{v}_m + \hat{u}_m) + (1-\alpha)\hat{w}_m \\ \hat{w}_{m+1} = \hat{A}_m \hat{w}_m + c \\ \hat{u}_{m+1} = \hat{A}_m \hat{u}_m \end{array} \right. \tag{6b}$$

Here, $\hat{w}_m$ and $\hat{u}_m$ are the same as in StocFP (4) and StocRB (5), respectively. $\hat{v}_{m+1}$ estimates the hyper-gradient (3) and is equal in expectation to $\hat{w}_m$ and $\hat{y}_m$. The update iteration for $\hat{v}_m$ is a slightly modified weighted average of $\hat{w}_m$ in (4b) and $\hat{y}_m$ in (5b). In fact, one can verify that $\hat{v}_{m+1} = \hat{w}_m$ when $\alpha = 0$ and $\hat{v}_{m+1} = \hat{y}_m$ when $\alpha = 1$. It is noteworthy that the update for $\hat{v}_{m+1}$ is not a simple weighted average of $\hat{w}_m$ and $\hat{y}_m$, because it uses the previously computed value $\hat{v}_m$ instead of $\hat{y}_m$. This recursive structure allows MixedFP to estimate Jacobian products using diverse sequences of Jacobian samples:

**Remark 2.** *When $\alpha \in (0,1)$ and given $m \in \{1, \ldots, M\}$, $\hat{v}_M$ uses $\hat{A}_{k-1} \cdots \hat{A}_{k-m}$ for any $k$ such that $m \le k \le M$ to estimate $A^m$. More specifically, with some $\sum_{k=m}^{M} a_{m,k-1} = 1$ such that $a_{m,k} > 0$*

---

Mixed Fixed-Point Iteration (**MixedFP**) (6)

$$\hat{v}_{M+1} = \quad c \qquad\qquad\qquad\qquad\qquad\qquad\qquad\qquad (= A^0 c)$$

$$+ \left( a_{1,M-1} \boxed{\hat{A}_{M-1}} + \cdots + a_{1,0} \boxed{\hat{A}_0} \right) c \qquad\qquad (\approx A^1 c)$$
$$\underbrace{\qquad\qquad\qquad\qquad\qquad\qquad}_{M \text{ terms}}$$

$$+ \left( a_{2,M-1} \boxed{\hat{A}_{M-1}\hat{A}_{M-2}} + \cdots + a_{2,1} \boxed{\hat{A}_1 \hat{A}_0} \right) c \qquad (\approx A^2 c)$$
$$\underbrace{\qquad\qquad\qquad\qquad\qquad\qquad}_{M-1 \text{ terms}}$$

$$\vdots$$

$$+ \left( a_{M-1,M-1} \boxed{\hat{A}_{M-1} \cdots \hat{A}_1} + a_{M-1,M-2} \boxed{\hat{A}_{M-2} \cdots \hat{A}_0} \right) c \qquad (\approx A^{M-1} c)$$

$$+ \boxed{\hat{A}_{M-1} \hat{A}_{M-2} \cdots \hat{A}_1 \hat{A}_0} \, c \qquad\qquad\qquad\qquad (\approx A^M c)$$

---

This diversity in the products of Jacobian samples cannot be obtained by simply taking a weighted average of $\hat{w}_m$ and $\hat{y}_m$. While MixedFP maintains the same order of computational complexity as StocFP and StocRB, it requires two vector-Jacobian-products ($\hat{A}_m \hat{w}_m$ and $\hat{A}_m \hat{u}_m$) at each step. This results in a higher per-iteration computational cost compared to StocFP and StocRB, which only requires a single JVP computation. However, our experimental results on computational cost in Appendix D.3 show that MixedFP remains competitive with existing methods even when evaluated on a wall-clock basis.

Under the following regularity conditions, we show that the expected error of MixedFP improves over StocFP and StocRB.

**Assumption 3.** *For every $\lambda \in \mathbb{R}^{d_\lambda}$, we assume that:*

(i) $\mathbb{E}[\|\hat{\varphi}(x, \lambda; \xi)\|^2] < \infty$ *for every $x \in \mathbb{R}^{d_x}$.*

(ii) *There exists a constant $\hat{\rho} < 1$ such that $\|\partial_x \hat{\varphi}(x, \lambda; \xi)\| \le \hat{\rho}$ for every $x \in \mathbb{R}^{d_x}$ and $\xi \in \mathcal{X}$.*

(iii) *The function $f(\cdot, \lambda)$ is Lipschitz continuous with some constant $L_f \ge 0$.*

| Method | Base estimator | KM iteration | Expected $\ell_2$ error bound |
|---|---|:---:|---|
| Grazzi et al. (2021) with $\eta = 1$ | StocFP | | $\sigma_1 + O(\rho_1^m)$ |
| Ji et al. (2021) | StocRB | | $\sigma_1 + O(\rho_1^m)$ |
| **MixedFP** | **StocFP & StocRB** | | $\frac{1+\alpha_1}{(1+\sqrt{\alpha_1})^2}\sigma_1 + O(\rho_1^m)$ |
| Grazzi et al. (2021) with constant $\eta$ | StocFP | ✓ | $\sigma_\eta + O(\rho_\eta^m)$ |
| **MixedFP-KM** with constant $\eta$ | **StocFP & StocRB** | ✓ | $\frac{1+\alpha_\eta}{(1+\sqrt{\alpha_\eta})^2}\sigma_\eta + O(\rho_\eta^m)$ |

Table 1: Comparison of the upper bound of the expected $\ell_2$ error in stochastic hypergradient estimation. Our MixedFP and MixedFP-KM can reduce non-decaying errors by $\frac{1+\alpha_\eta}{(1+\sqrt{\alpha_\eta})^2} < 1$ times smaller than their counterparts. All symbols are defined in Theorems 4.1 and 4.5 and we assume $\alpha \leq \hat{\rho}$ in this table.

**Theorem 4.1** (MixedFP). *Suppose Assumptions 1 to 3 hold and $\alpha \in [0,1]$, then for any $m \geq 0$,*

$$\mathbb{E}\left[\|\hat{v}_m - \mathrm{d}_x f\|^2\right] \leq \begin{cases} \sigma_1 + O\left(\rho_1^m\right) & \text{if } \alpha \in \{0,1\}, \\ \dfrac{1+\alpha_1}{(1+\sqrt{\alpha_1})^2}\sigma_1 + O\left(\max\{\rho_1,\alpha_1\}^m\right) & \text{otherwise,} \end{cases}$$

*where*

$$\sigma_1 = \frac{2L_f^2\hat{\rho}^2}{(1-\hat{\rho})^2(1-\rho^2)}, \quad \rho_1 = \hat{\rho}^2, \quad \alpha_1 = \alpha^2.$$

This result shows that the scale of the non-decaying term, $\frac{1+\alpha_1}{(1+\sqrt{\alpha_1})^2} \leq 1$, becomes smaller as $\alpha$ increases, but when $\alpha$ is too large, more specifically when $\alpha > \hat{\rho}$, the decay rate becomes slower than $O(\rho_1^m)$. Recalling that $\alpha \in \{0,1\}$ corresponds to StocFP and StocRB, this indicates that $\alpha$ chosen in $0 < \alpha < \hat{\rho}$ improves the non-decaying term and the decaying term over these conventional methods. Table 1 summarizes these findings. This analysis also suggests that for sufficiently large $m$, the smallest error is achieved when $\alpha$ is infinitely close to, but strictly less than, one. However, for finite $m$, a smaller $\alpha$ may be optimal if accelerating the decaying term outweighs the reduction of the non-decaying term. These observations align with the empirical results in Section 5.1.

## 4.2 MixedFP-KM: Application of the Stochastic Krasnosel'skiĭ-Mann (KM) Iteration

This section shows that the stochastic KM iteration can be applied to MixedFP, enabling almost sure convergence to the true hypergradient.

The stochastic KM iteration (Grazzi et al., 2021; Bravo & Cominetti, 2024) is an algorithm that finds the fixed point of a contraction mapping using its unbiased estimation:

**Theorem 4.2** (Stochastic KM iteration (Grazzi et al., 2021)). *Let $\zeta$ be a random variable with values in a measurable space $\mathcal{Z}$ and $(\zeta_m)_{m\in\mathbb{N}}$ be independent copies of $\zeta$. Let $T : \mathbb{R}^d \to \mathbb{R}^d$ be a contraction mapping and suppose a mapping $\hat{T} : \mathbb{R}^d \times \mathcal{Z} \to \mathbb{R}^d$ satisfies $\mathbb{E}[\hat{T}(s;\zeta)] = T(s)$ and $\mathbb{V}[\hat{T}(s;\zeta)] \leq \varsigma_1 + \varsigma_2\|T(s) - s\|$ for any $s \in \mathbb{R}^d$. Consider the stochastic Krasnosel'skiĭ-Mann iteration defined by*

$$s_{m+1} = (1 - \eta_m)s_m + \eta_m\hat{T}(s_m;\zeta_m), \tag{7}$$

*where stepsizes $(\eta_m)_{m\in\mathbb{N}}$ satisfy*

$$\eta_k \leq \frac{1}{1+\varsigma_2}, \ \forall k \in \mathbb{N}, \qquad \sum_{k=0}^{\infty}\eta_k = \infty, \qquad \sum_{k=0}^{\infty}\eta_k^2 < \infty. \tag{8}$$

*Then the sequence $(s_m)_{m\in\mathbb{N}}$ converges almost surely to the fixed point of $T$.*

Next, we show that MixedFP is a contraction mapping. To do this, we reformat MixedFP into the following mapping of a concatenated vector.

$$\hat{F}(z;\xi) = \hat{B}z + d, \tag{9}$$

$$\text{where} \quad z = \begin{bmatrix} v \\ w \\ u \end{bmatrix} \in \mathbb{R}^{3d_x}, \quad \hat{B} = \begin{bmatrix} \alpha I & \alpha I & (1-\alpha)I \\ O & \hat{A} & O \\ O & O & \hat{A} \end{bmatrix} \in \mathbb{R}^{3d_x \times 3d_x}, \quad d = \begin{bmatrix} 0 \\ c \\ 0 \end{bmatrix} \in \mathbb{R}^{3d_x}.$$

The mapping $\hat{F}$ has the following favorable properties that enable the application of the stochastic KM iteration.

**Lemma 4.3.** *When Assumptions 1 and 2 hold, the function $F(z) = \mathbb{E}[\hat{F}(z;\xi)]$ with $\alpha \in [0,1)$ is a contraction mapping that has the unique fixed point $[\mathrm{d}_x f^\top \ \mathrm{d}_x f^\top \ 0^\top] \in \mathbb{R}^{3d_x}$.*

By applying (7) to the mapping $\hat{F}$ in (9), we obtain the following algorithm, which we name MixedFP-KM.

$$v_0 = 0 \in \mathbb{R}^{d_x}, \ w_0 = u_0 = c$$

$$\text{For } m = 0, \ldots, M-1:$$

$$\left| \begin{array}{l} v_{m+1} = (1-\eta_m)v_m + \eta_m(\alpha(v_m + u_m) + (1-\alpha)w_m) \\ w_{m+1} = (1-\eta_m)w_m + \eta_m(\hat{A}_m w_m + c) \\ u_{m+1} = (1-\eta_m)u_m + \eta_m \hat{A}_m u_m \end{array} \right.$$

From Theorem 4.2 and Lemma 4.3, it follows that $v_m$ converges to $\mathrm{d}_x f$ with appropriate scheduling of $\eta_m$.

**Theorem 4.4.** *Let $\alpha \in [0,1)$, $q = \max\{\alpha, \rho\}$, and $\hat{q} = \max\{\alpha, \hat{\rho}\}$. When Assumptions 1 to 3 hold and $\eta_m$ satisfies (8) with $\varsigma_2 = 2(\hat{q}^2 + q^2)/(1-q)^2$, then*

$$\lim_{m \to \infty} v_m = \mathrm{d}_x f \quad a.s.$$

For fixed stepsizes, the error can be bounded by an exponentially decaying term and a non-decaying term.

**Theorem 4.5** (MixedFP-KM). *Let $\alpha \in [0,1)$. When $\eta_m = \eta < \frac{1}{1-\hat{\rho}}$ and Assumptions 1 to 3 hold, then for any $m \geq 0$,*

$$\mathbb{E}\left[\|v_m - \mathrm{d}_x f\|^2\right] \leq \frac{1 + \alpha_\eta}{\left(1 + \sqrt{\alpha_\eta}\right)^2}\sigma_\eta + O\left(\max\{\rho_\eta, \alpha_\eta\}^m\right),$$

*where*

$$\sigma_\eta = \frac{2\eta L_f^2 \hat{\rho}^2}{(1-\hat{\rho})^2(2 - \eta(1-\rho))(1-\rho)}, \quad \rho_\eta = (1 - \eta + \eta\hat{\rho})^2, \quad \alpha_\eta = (1 - \eta + \eta\alpha)^2.$$

For comparison, we show our result for the existing method (Grazzi et al., 2021), which applies the stochastic KM iteration to StocFP[2].

**Theorem 4.6** (Stochastic KM iteration of StocFP). *Suppose $\eta_m = \eta < \frac{1}{1-\hat{\rho}}$ and Assumptions 1 to 3 hold. Then for any $m \geq 0$,*

$$\mathbb{E}\left[\|w_m - \mathrm{d}_x f\|^2\right] \leq \sigma_\eta + O\left(\rho_\eta^m\right),$$

*where $\sigma_\eta$ and $\rho_\eta$ are as defined in Theorem 4.5.*

The non-decaying terms in both Theorems 4.5 and 4.6 can be made arbitrarily small by the choice of $\eta$ at the cost of a slow decay rate. The observations obtained from comparing these two theorems are essentially the same as those obtained in Theorem 4.1. That is, when $0 < \alpha < \hat{\rho}$, both the non-decaying and decaying term improve over the existing method (Grazzi et al., 2021). This was also empirically confirmed in Section 5.2.

We note that our theoretical analysis for decreasing stepsizes (Theorem 4.4) provides only asymptotic convergence guarantees without non-asymptotic error bounds. While our experiments employ decreasing stepsizes, demonstrating its practical performance, the finite-time performance analysis is limited to the fixed stepsize case (Theorem 4.5). Future work could focus on extending our theoretical analysis to provide non-asymptotic error bounds for decreasing stepsizes.

---

[2]Theorem 4.6 differs from a result obtained from Grazzi et al. (2021, Theorem 4.2). As discussed in Appendix B.2, this is solely for a fair comparison with our method, and in some cases Theorem 4.6 is even tighter than the original bound.

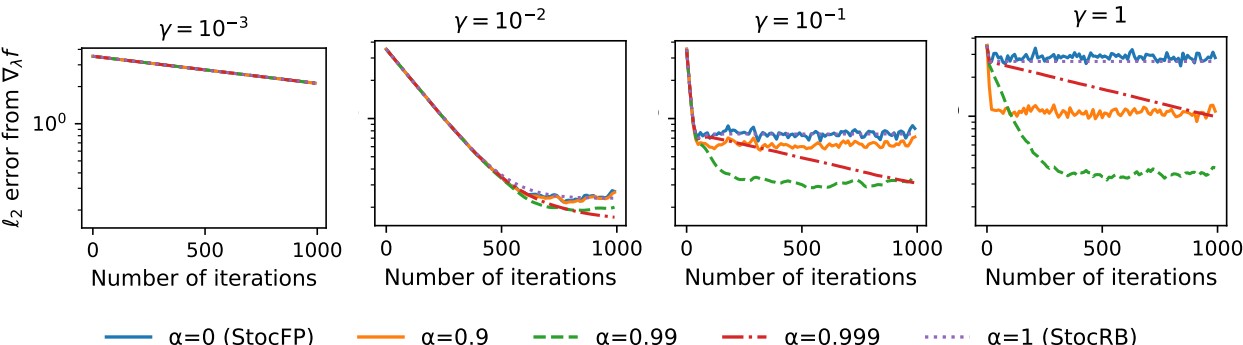

Figure 1: Mean L2 error of hypergradients estimated by MixedFP using different values of $\alpha$. We adopted a synthetic inner-problem $\varphi(x(\lambda), \lambda) = (I - \gamma\hat{H})x(\lambda) + B\lambda$ and plotted results obtained with different $\gamma$.

## 5 Experiments

In this section, we first investigate the relationship between the newly introduced parameter $\alpha$ and the convergence properties of MixedFP. Then, we compare the hypergradient estimation accuracy of our proposed method with existing approaches in various real-world task settings.

### 5.1 Effect of Mixing Rate

#### 5.1.1 Settings

This experiment evaluates the hypergradient estimation error in a synthetic setting where we can explicitly control the Jacobian matrix samples. Specifically, we compute the hypergradient for the following case:

$$f(x(\lambda), \lambda) = c^\top x(\lambda) + d^\top \lambda, \qquad \varphi(x, \lambda) = (I - \gamma\hat{H})x + B\lambda,$$

where $\hat{H} \in \mathbb{R}^{d_x \times d_x}$ is a random variable sampled from a discrete uniform distribution $\hat{H} \sim$ Uniform($\{H_1, \ldots, H_n\}$). Each matrix in $H_1, \ldots, H_n$ was constructed such that its eigenvalues follow the uniform distribution over $[0, 1-\epsilon]$ with a small constant $\epsilon \in \mathbb{R}_{++}$ to meet our assumptions. We tested different $\gamma$ over $(0, 1]$, which corresponds to varying the values of $\rho$ and $\hat{\rho}$. The other coefficients $c \in \mathbb{R}^{d_x}$, $d \in \mathbb{R}^{d_\lambda}$, and $B \in \mathbb{R}^{d_x \times d_\lambda}$ were generated by sampling their elements independently from a uniform distribution over $[0, 1]$. Note that $x$ and $\lambda$ vanish upon differentiation, thus their values do not affect the hypergradient estimation. The experiment was run 100 times with different seed values for sampling $\hat{H}$.

#### 5.1.2 Results and Discussion

Fig. 1 shows the mean squared error of hypergradients at different $\gamma$ and $\alpha$.

Estimations with $\alpha = 0.99$ or $\alpha = 0.999$ consistently outperform both $\alpha = 0$ (StocFP) and $\alpha = 1$ (StocRB) across different values of $\gamma$, highlighting the benefit of our mixing strategy. The results suggest that the optimal value of $\alpha$ is close to but not exactly one, aligning with the observation from Theorem 4.1. The superior performance of $\alpha = 0.99$ over the larger $\alpha = 0.999$ is also consistent with Theorem 4.1: for finite iterations, a slightly smaller $\alpha$ can be optimal when faster decay outweighs the increased non-decaying term.

Additionally, we observe that increasing $\gamma$ accelerates the decay rate while enlarging the non-decay term. Higher values of $\gamma$ result in smaller values for both $\rho$ and $\hat{\rho}$, which aligns with a theoretical finding in Theorem 4.1 that implies a smaller $\hat{\rho}$ leads to a larger non-decaying term $\sigma_1$ and a faster decay rate $\rho_1$.

Furthermore, some settings show interesting patterns in their error curves. When $\gamma = 10^{-1}$ with $\alpha = 0.999$ or $0.99$, the error seems to decay at two distinct rates: initially faster, then slower after a certain number of iterations (around $m = 50$). This seems to reflect the bound $O(\max\{\rho_1, \alpha_1\}^m)$, which is yielded from the sum of two terms with different decay rates $\rho_1$ and $\alpha_1$ (as in Appendix A.5 (39)); the term with the faster decay rate converges quickly, and for sufficiently large $m$, the slower decay rate eventually dominates.

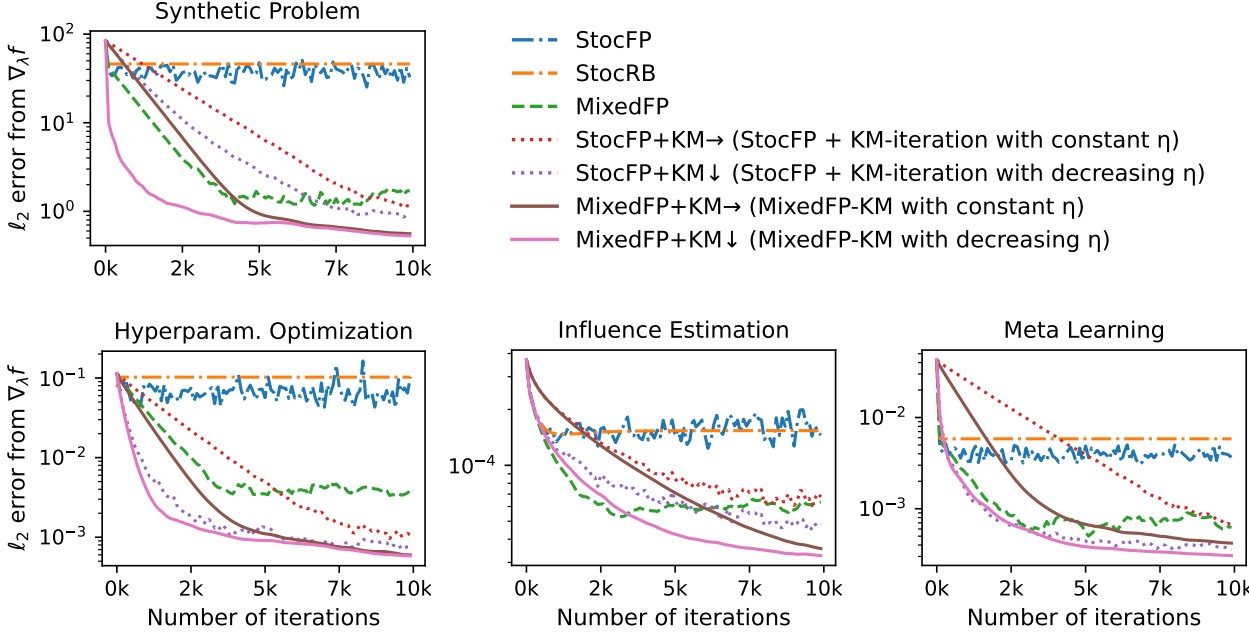

Figure 2: Comparison of stochastic hypergradient estimation methods evaluated on different tasks. Each line shows a mean squared error over 10 trials with different Jacobian sample sequences $(\hat{A}_m)_{m \in \mathbb{N}}$. Method-specific parameters were optimized by grid search, selecting settings with minimal average error over the last 1,000 iterations.

## 5.2 Comparison with Existing Approaches

In this section, we consider several real-world machine learning settings to compare the hypergradient estimation accuracy of our proposed method with existing approaches across various combinations of the dataset, inner-problem, and outer-cost.

### 5.2.1 Tasks

We evaluated the performance of hypergradient estimation methods for the following task settings:

- **Synthetic problem** evaluates the estimation accuracy of hypergradients of the synthetic outer-cost with the synthetic inner-problem used in Section 5.1.

- **Hyperparameter optimization** performs binary classification on Adult Income dataset (Becker & Kohavi, 1996), where the outer-parameter is a vector of regularization coefficients, each corresponding to a dimension of the inner-parameter.

- **Influence estimation** conducts multi-class classification on Fashion-MNIST (Xiao et al., 2017), where the outer-parameter is a vector of loss masks, each assigned to a corresponding sample.

- **Meta learning** tackles a regression problem on the California Housing dataset (Pace & Barry, 1997), where the outer-parameter determines a biased regularization term.

Except for the synthetic setting, we adopt $\hat{\varphi}(x, \lambda; \xi) = x - \gamma \partial_x g(x, \lambda; \xi)$ and define $f$ and $g$ for each setting as follows. We denote training inputs and outputs by $\xi_{\text{in}}$ and $\xi_{\text{out}}$, and use $(\xi'_{\text{in}}, \xi'_{\text{out}}) \in \Xi_{\text{val}}$ for the independent validation dataset.

**Hyperparameter optimization** refers to the hypergradient estimation problem in a bilevel problem where the regularization coefficients serve as hyperparameters. Specifically, as in Grazzi et al. (2021), we compute

the hypergradient for an outer-problem that finds the optimal hyperparameters $\lambda$ whose elements are different regularization coefficients for each dimension of $x$:

$$f(x, \lambda) = \sum_{(\xi'_{\text{in}}, \xi'_{\text{out}}) \in \Xi_{\text{val}}} \text{BCE}(\xi'_{\text{out}}, \xi'^{\top}_{\text{in}} x), \qquad g(x, \lambda; (\xi_{\text{in}}, \xi_{\text{out}})) = \text{BCE}(\xi_{\text{out}}, \xi^{\top}_{\text{in}} x) + \frac{1}{2} x^{\top} \text{diag}(\lambda) x, \quad (10)$$

where $\xi_{\text{in}} \in \mathbb{R}^{14}$, $\xi_{\text{out}} \in \{0, 1\}$, $\text{BCE}(p, q) = \log(1 + \exp(-pq))$, and $\text{diag}(\lambda)$ denotes the diagonal matrix whose diagonal entries are the elements of $\lambda$.

**Influence estimation** (Koh & Liang, 2017; Khanna et al., 2019) estimates the change in performance when a sample is excluded from the inner-problem. Influence estimation is equivalent to estimating the hypergradient of an outer-parameter that determines the loss mask of each sample. That is, with the sample index $j = \text{Uniform}(\{1, \dots, N\})$ with dataset size $N$,

$$f(x, \lambda) = \sum_{(\xi'_{\text{in}}, \xi'_{\text{out}}) \in \Xi_{\text{val}}} \text{CE}(\xi'_{\text{out}}, \xi'^{\top}_{\text{in}} x), \qquad g(x, \lambda; j) = \lambda_j \text{CE}(\xi_{\text{out},j}, \xi^{\top}_{\text{in},j} x), \quad (11)$$

where $\text{CE}(p, q) = -\sum_{c=1}^{10} p_c \log\left(\exp(q_c) / \sum_{c'=1}^{10} \exp(q_{c'})\right)$ and $(\xi_{\text{in},j}, \xi_{\text{out},j}) \in \mathbb{R}^{784} \times \{0, 1\}^{10}$ denotes the $j$-th sample in the training dataset. For the values of the loss mask, let $\mathbf{1} \in \{1\}^N$ be the ones vector and $\mathbf{1}_{\neq j} \in \{0, 1\}^N$ denote a vector with ones except for a zero at the $j$-th index. Then, we can express the inner-solution obtained *including* or *excluding* the $j$-th sample as $x(\mathbf{1})$ or $x(\mathbf{1}_{\neq j})$, respectively. The linear approximation of the change in $f$ when the $j$-th sample is excluded from the training dataset is given by

$$f(x(\mathbf{1}_{\neq j}), \mathbf{1}_{\neq j}) - f(x(\mathbf{1}), \mathbf{1}) \approx \text{d}_{\lambda} f(x(\lambda), \lambda)\big|^{\top}_{\lambda=\mathbf{1}} (\mathbf{1}_{\neq j} - \mathbf{1}) = -\text{d}_{\lambda_j} f(x(\lambda), \lambda)\big|_{\lambda=\mathbf{1}} \in \mathbb{R}.$$

By computing a vector $-\text{d}_{\lambda} f(x(\lambda), \lambda)\big|_{\lambda=\mathbf{1}} \in \mathbb{R}^N$, we can obtain this approximation for all samples in the dataset. Thus, influence estimation reduces to estimation of the hypergradient.

Further details on influence estimation and its connections to existing work (Koh & Liang, 2017) are provided in Appendix D.2.2.

**Meta learning** is the following task adopted in Grazzi et al. (2020), which simplifies the task called implicit meta learning (Denevi et al., 2019; Rajeswaran et al., 2019).

$$f(x, \lambda) = \frac{1}{2} \sum_{(\xi'_{\text{in}}, \xi'_{\text{out}}) \in \Xi_{\text{val}}} \|\xi'^{\top}_{\text{in}} x(\lambda) - \xi'_{\text{out}}\|^2, \qquad g(x, \lambda; (\xi_{\text{in}}, \xi_{\text{out}})) = \frac{1}{2} \|\xi^{\top}_{\text{in}} x - \xi_{\text{out}}\|^2 + \frac{\mu}{2} \|x - \lambda\|^2, \quad (12)$$

where $(\xi_{\text{in}}, \xi_{\text{out}}) \in \mathbb{R}^8 \times \mathbb{R}$, $\mu \in \mathbb{R}_{++}$, and $\lambda \in \mathbb{R}^{d_x}$.

### 5.2.2 Baselines and Procedure

Hypergradient estimation is performed using the five methods: `StocFP`, `StocRB`, `MixedFP`, StocFP with KM-iteration (denoted as `StocFP+KM→` and `StocFP+KM↓`), and MixedFP with KM-iteration (denoted as `MixedFP+KM→` and `MixedFP+KM↓`). KM→ and KM↓ represent constant and linearly decreasing $\eta_m$, respectively, and we used $\eta_m = \beta/(\delta + m)$ as the decreasing stepsize following Grazzi et al. (2021).

For each setting and method, we conducted the experiment 10 times using different seed values to vary the sequence $(\hat{A}_m)_{m < M}$, and calculated the mean squared error $\ell_2$ with respect to $\text{d}_x f$. We computed the exact values of $\partial_x f$, $\partial_\lambda f$, and $\partial_\lambda \varphi$ in (2) to ensure that the only source of randomness came from $(\hat{A}_m)_{m < M}$. We approximated $x(\lambda)$ as accurately as possible by minimizing $g$ using full-batch optimizations until they converge. The impact of approximation errors in $x(\lambda)$ is analyzed only theoretically in Appendix C. For the outer-parameter $\lambda$, we used task-specific initialization values without performing outer-optimization.

To ensure a fair comparison between the baseline and proposed methods, we use the parameters that yield the best performance for each method. These parameters include $\eta, \alpha, \beta$ and $\delta$, which were determined vigrida search by minimizing the average error over the last 1,000 steps.

More detailed settings are deferred to our appendix.

### 5.2.3   Results and Discussion

In all tasks, `MixedFP+KM↓` yielded the best performance (Table 1). Our theoretical analysis quantifies the improvements achieved under a fixed stepsize (by comparing Theorem 4.5 and Theorem 4.6), while for a decreasing stepsize, we have only established asymptotic convergence (Theorem 4.4). Nevertheless, these results demonstrate that, in practice, MixedKM can also improve estimation accuracy with decreasing stepsizes. When using KM-iteration with constant stepsizes, `MixedFP+KM→` achieves lower error than its counterpart `StocFP-KM→`, which supports our results of Theorem 4.5. Among methods without KM-iteration, `StocFP` and `StocRB` tend to reach plateaus in the early iterations, while `MixedFP` continues to reduce error over a relatively larger number of steps.

## 6   Conclusion

We presented a stochastic hypergradient estimation method that reduces variance incurred by estimation error of the product of the inner-iteration's Jacobian matrices. Our proposed MixedFP combines existing StocFP and StocRB estimators to leverage their distinct sequences of Jacobian samples, thereby reducing estimation variance without increasing computational complexity. Moreover, by applying the stochastic KM iteration into MixedFP, we established almost sure convergence to the true hypergradient. Empirical evaluations on synthetic and real-world tasks, including hyperparameter optimization, data influence estimation, and meta learning, validated our theoretical findings and demonstrated improved estimation accuracy in practical scenarios.

Future work will focus on extending our analysis to the case of decreasing stepsizes and exploring efficiency improvements in the application, including bilevel optimization.

## Acknowledgements

This work was supported in part by Grant-in-Aid for Scientific Research(B) 23K28146.

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

# A    Theoretical Analysis of MixedFP-KM

In this section, we provide a detailed analysis of the proposed MixedFP-KM. We first recall the iteration schemes, then present key lemmas giving explicit forms of the iterates and the corresponding error bounds.

## A.1    Recap of the Iterations

Let $\eta \in \mathbb{R}_{++}$ be a stepsize. We denote by $(v_m)$, $(w_m)$, and $(u_m)$ the sequences generated by the following iteration:

$$v_0 = 0, \tag{13}$$
$$w_0 = c, \tag{14}$$
$$u_0 = c, \tag{15}$$

and for $m = 0, 1, 2, \ldots$

$$v_{m+1} = (1 - \eta)v_m + \eta \left(\alpha \left(u_m + v_m\right) + \overline{\alpha} w_m\right) = r v_m + \eta \left(\alpha u_m + \overline{\alpha} w_m\right), \tag{16}$$
$$w_{m+1} = (1 - \eta)w_m + \eta \left(\hat{A}_m w_m + c\right) = \hat{P}_m w_m + \eta c, \tag{17}$$
$$u_{m+1} = (1 - \eta)u_m + \eta \left(\hat{A}_m u_m\right) = \hat{P}_m u_m, \tag{18}$$

where $\alpha \in [0, 1]$, $\overline{\alpha} = 1 - \alpha$, $r = 1 - \eta \left(1 - \alpha\right)$, and $\hat{P}_m = I - \eta \left(I - \hat{A}_m\right)$.

## A.2    Proof of Asymptotic Convergence of MixedFP-KM

This section provides the proofs for Lemma 4.3 and Theorem 4.4.

**Lemma 4.3.** *When Assumptions 1 and 2 hold, the function $F(z) = \mathbb{E}[\hat{F}(z; \xi)]$ with $\alpha \in [0, 1)$ is a contraction mapping that has the unique fixed point $[\mathrm{d}_x f^\top \ \mathrm{d}_x f^\top \ 0^\top] \in \mathbb{R}^{3d_x}$.*

*Proof.* Let $B = \mathbb{E}[\hat{B}]$. Because $B$ is a block triangle matrix, its eigenvalues are those of diagonal blocks, that are ensured to lie in $[0, 1)$ by Assumption 1 and $\alpha \in [0, 1)$. Hence, $F$ is a contraction mapping which has a unique fixed point.

Let the first, second, and third blocks of the fixed point $z^* = T(z^*)$ be $v^*$, $w^*$, and $u^*$. From Assumption 1, we know that $u^* = 0$. Under Assumption 2, $w^*$ satisfies $w^* = Aw^* + c$ thus $w^* = (I - A)^{-1}c = \mathrm{d}_x f$. Finally, since we know that $v^* = \alpha v^* + (1 - \alpha)\mathrm{d}_x f$, which is true only when $v^* = \mathrm{d}_x f$ since $1 - \alpha \neq 0$. □

**Theorem 4.4.** *Let $\alpha \in [0, 1)$, $q = \max\{\alpha, \rho\}$, and $\hat{q} = \max\{\alpha, \hat{\rho}\}$. When Assumptions 1 to 3 hold and $\eta_m$ satisfies (8) with $\varsigma_2 = 2(\hat{q}^2 + q^2)/(1 - q)^2$, then*

$$\lim_{m \to \infty} v_m = \mathrm{d}_x f \quad a.s.$$

*Proof.* Let $q = \|B\|$ and $\hat{q} = \|\hat{B}\|$, where $B = \mathbb{E}[\hat{B}]$. Since $B$ is a block triangular matrix, its eigenvalues are determined by those of its diagonal blocks, giving us $q = \max\{\alpha, \rho\}$ from Assumption 1. Similarly, we can establish that $\hat{q} = \max\{\alpha, \hat{\rho}\}$ under Assumption 3. Applying Theorem 4.3 from Grazzi et al. (2021), we obtain $\varsigma_2 = 2\frac{\hat{q}^2 + q^2}{(1 - q)}$. Therefore, combining Theorem 4.2 and Lemma 4.3, we conclude that the statement holds true. □

## A.3    Closed-Form Expressions of MixedFP-KM

We first derive closed-form expressions (or general forms) for the iterates $(v_m), (w_m)$, and $(u_m)$ under the MixedFP-KM iteration (16) to (18) with a fixed step-size $\eta$. Note that in this section, we exclusively consider the case where the step-size remains constant throughout the iterations to avoid any confusion.

**Lemma A.1** (General-Term Representation). *Under the MixedFP-KM iteration* (13) *to* (18) *with a fixed step-size* $\eta = \eta_m$, *for* $m \geq 0$ *we have:*

$$v_{m+1} = \eta \sum_{k=0}^{m} r^{m-k} \left( \alpha u_k + \overline{\alpha} w_k \right), \tag{19}$$

$$w_{m+1} = \left( \prod_{j=0}^{m} \hat{P}_j \right) c + \eta \sum_{k=0}^{m} \left( \prod_{j=k+1}^{m} \hat{P}_j \right) c, \tag{20}$$

$$u_{m+1} = \left( \prod_{j=0}^{m} \hat{P}_j \right) c, \tag{21}$$

*where* $\alpha \in [0,1]$, $\overline{\alpha} = 1 - \alpha$, $r = 1 - \eta (1 - \alpha)$, *and* $\hat{P}_m = I - \eta \left( I - \hat{A}_m \right)$.

*Proof.* We prove each identity by induction.

(Proof of (19)) For $m = 0$, (16) matches (19) since

$$v_1 = \eta r^0 \left( \alpha u_0 + \overline{\alpha} w_0 \right).$$

Assume (19) holds for some $m \geq 0$. Then from (16),

$$
\begin{aligned}
v_{m+2} &= r v_{m+1} + \eta \left( \alpha u_{m+1} + \overline{\alpha} w_{m+1} \right) \\
&= r \left( \eta \sum_{k=0}^{m} r^{m-k} \left( \alpha u_k + \overline{\alpha} w_k \right) \right) + \eta \left( \alpha u_{m+1} + \overline{\alpha} w_{m+1} \right) \\
&= \eta \sum_{k=0}^{m+1} r^{m+1-k} \left( \alpha u_k + \overline{\alpha} w_k \right).
\end{aligned}
$$

This completes the inductive step.

(Proof of (20)) For $m = 0$, from (17), we have

$$w_1 = \hat{P}_0 c + \eta c,$$

since $w_0 = c$. This is consistent with (20) for $m = 0$ because

$$w_1 = \left( \prod_{j=0}^{0} \hat{P}_j \right) c + \eta \sum_{k=0}^{0} \left( \prod_{j=k+1}^{0} \hat{P}_j \right) c = \hat{P}_0 c + \eta c.$$

Assume (20) holds for some $m \geq 0$. Then from (17),

$$
\begin{aligned}
w_{m+2} &= \hat{P}_{m+1} w_{m+1} + \eta c \\
&= \hat{P}_{m+1} \left( \prod_{j=0}^{m} \hat{P}_j \right) c + \hat{P}_{m+1} \eta \sum_{k=0}^{m} \left( \prod_{j=k+1}^{m} \hat{P}_j \right) c + \eta c \\
&= \left( \prod_{j=0}^{m+1} \hat{P}_j \right) c + \eta \sum_{k=0}^{m+1} \left( \prod_{j=k+1}^{m+1} \hat{P}_j \right) c.
\end{aligned}
$$

This completes the proof for $w_{m+1}$.

(Proof of (21)) The derivation is analogous to that of $w_m$, except that no extra $\eta c$ term added. For $m = 0$, (21) and (18) are consistent because both are

$$u_1 = \hat{P}_0 c.$$

from $u_0 = c$.

The inductive step is essentially the same as above, giving

$$u_{m+2} = \hat{P}_{m+1} u_{m+1} = \left(\prod_{j=0}^{m+1} \hat{P}_j\right) c.$$

Hence (21) holds. □

### A.4 General Error Terms

This section considers differences between $(v_m, w_m, u_m)$ and their deterministic counterparts $(\overline{v}_m, \overline{w}_m, \overline{u}_m)$, namely,

$$\overline{v}_0 = 0, \tag{22}$$
$$\overline{w}_0 = c, \tag{23}$$
$$\overline{u}_0 = c, \tag{24}$$

and for $m = 0, 1, 2, \ldots$

$$\overline{v}_{m+1} = (1-\eta)\overline{v}_m + \eta\left(\alpha\left(\overline{u}_m + \overline{v}_m\right) + \overline{\alpha}\overline{w}_m\right) = r\overline{v}_m + \eta\left(\alpha\overline{u}_m + \overline{\alpha}\overline{w}_m\right), \tag{25}$$
$$\overline{w}_{m+1} = (1-\eta)\overline{w}_m + \eta\left(A\overline{w}_m + c\right) = P_m\overline{w}_m + \eta c, \tag{26}$$
$$\overline{u}_{m+1} = (1-\eta)\overline{u}_m + \eta\left(A\overline{u}_m\right) = P_m\overline{u}_m, \tag{27}$$

where $P = I - \eta(I - A)$. The next lemma gives a useful expression for these differences.

**Lemma A.2** (General Error Terms). *Under the MixedFP-KM iteration* (13) *to* (18) *with* $\eta_m = \eta$ *and its deterministic counterpart* (22) *to* (27), *for* $s \geq 1$ *we have,*

$$v_s - \overline{v}_s = \eta \sum_{k=0}^{s-1} r^{s-1-k} \left(\alpha\left(u_k - \overline{u}_k\right) + \overline{\alpha}\left(w_k - \overline{w}_k\right)\right), \tag{28}$$

$$w_s - \overline{w}_s = \eta \sum_{j=0}^{s-1} P^{s-1-j}(\hat{A}_j - A)\left\{\left(\prod_{i=0}^{j-1}\hat{P}_i\right)c + \eta\sum_{k=0}^{j-1}\left(\prod_{i=k+1}^{j-1}\hat{P}_i\right)c\right\}, \tag{29}$$

$$u_s - \overline{u}_s = \eta \sum_{k=0}^{s-1} P^{s-1-k}\left(\hat{A}_k - A\right)\left(\prod_{i=0}^{k-1}\hat{P}_i\right)c, \tag{30}$$

*where* $P = I - \eta(I - A)$.

*Proof.* We give the argument for each of the three differences, using induction and the results in Lemma A.1.

(Difference for $v_m$) As the direct consequence of Lemma A.1, the general term of the deterministic case is given by

$$\overline{v}_{s+1} = \eta \sum_{k=0}^{s} r^{s-k}\left(\alpha\overline{u}_k + \overline{\alpha}\overline{w}_k\right), \tag{31}$$

One can directly obtain (28) by comparing the general terms (19) and (31).

(Difference for $w_s$) By subtracting (26) from (17), the following recursion is shown to hold:

$$\begin{aligned}
w_{s+1} - \overline{w}_{s+1} &= \hat{P}_s w_s - P\overline{w}_s + \eta c - \eta c \\
&= P\left(w_s - \overline{w}_s\right) + (\hat{P}_s - P)w_s \\
&= P\left(w_s - \overline{w}_s\right) + \eta(\hat{A}_s - A)w_s. \tag{32}
\end{aligned}$$

Hence, (29) is consistent with (32) when $s = 0$, because both are

$$w_1 - \overline{w}_1 = \eta(\hat{A}_0 - A)c, \tag{33}$$

using the fact that $w_0 = \overline{w}_0 = c$. Assume (29) holds for some $s \geq 1$. Then, from (32) and (20),

$$
\begin{aligned}
w_{s+1} - \overline{w}_{s+1} &= P\left(w_s - \overline{w}_s\right) + \eta(\hat{A}_s - A)w_s \\
&= \eta P \sum_{j=0}^{s-1} P^{s-1-j}(\hat{A}_j - A)\left\{\left(\prod_{i=0}^{j-1}\hat{P}_i\right)c + \eta\sum_{k=0}^{j-1}\left(\prod_{i=k+1}^{j-1}\hat{P}_i\right)c\right\} \\
&\quad + \eta(\hat{A}_s - A)\left\{\left(\prod_{i=0}^{s-1}\hat{P}_i\right)c + \eta\sum_{k=0}^{s-1}\left(\prod_{i=k+1}^{s-1}\hat{P}_i\right)c\right\} \\
&= \eta\sum_{j=0}^{s} P^{s-j}(\hat{A}_j - A)\left\{\left(\prod_{i=0}^{j-1}\hat{P}_i\right)c + \eta\sum_{k=0}^{j-1}\left(\prod_{i=k+1}^{j-1}\hat{P}_i\right)c\right\}
\end{aligned}
$$

This completes the proof for $s + 1$.

(Difference for $u_s$) Similar steps yield (30), now without the extra $c$ term in the iteration for $u_{m+1}$. Namely, substraction of (18) from (27) yields

$$u_{s+1} - \overline{u}_{s+1} = P\left(u_s - \overline{u}_s\right) + \eta(\hat{A}_s - A)u_s,$$

and one can show that this is consistent with (30) when $s = 0$. When (30) is assumed to be true for some $s \geq 1$, then (30) holds true for $s + 1$ because

$$u_{s+1} - \overline{u}_{s+1} = P\left(u_s - \overline{u}_s\right) + \eta(\hat{A}_s - A)u_s \tag{34}$$

$$= \eta P \sum_{j=0}^{s-1} P^{s-1-j}(\hat{A}_j - A)\left(\prod_{i=0}^{j-1}\hat{P}_i\right)c + \eta(\hat{A}_s - A)\left(\prod_{i=0}^{s-1}\hat{P}_i\right)c \tag{35}$$

$$= \eta\sum_{j=0}^{s} P^{s-j}(\hat{A}_j - A)\left(\prod_{i=0}^{j-1}\hat{P}_i\right)c. \tag{36}$$

$\square$

## A.5 Bounding Variance Norm of MixedFP-KM

We prove the bound on $\mathbb{E}\left[\|v_m - \overline{v}_m\|^2\right]$ starting with proving a useful lamma.

**Lemma A.3.** *Let $x > 1$ and let $m \geq 0$ be an integer. Then*

$$\sum_{k=0}^{m-1}(2(m-k)-1)x^k \leq \frac{x^m(x+1)}{(1-x)^2}.$$

*Proof.* First, observe that

$$\sum_{k=0}^{m-1}(2(m-k)-1)x^k = (2m-1)\sum_{k=0}^{m-1}x^k - 2\sum_{k=0}^{m-1}kx^k.$$

Since $x \neq 1$, the geometric series partial sum is

$$\sum_{k=0}^{m-1}x^k = \frac{1-x^m}{1-x}.$$

A standard result (or by differentiating the geometric series) gives

$$\sum_{k=0}^{m-1} kx^k = \frac{x - mx^m + (m-1)x^{m+1}}{(1-x)^2},$$

Multiplying the desired inequality

$$\sum_{k=0}^{m-1} \left(2(m-k) - 1\right) x^k \leq \frac{x^m(x+1)}{(1-x)^2}$$

by $(1-x)^2$ (which is positive for $x > 1$) reduces the task to proving

$$(2m - 1)(1 - x^m)(1 - x) - 2\left(x - mx^m + (m-1)x^{m+1}\right) \leq x^m(x+1).$$

A short computation shows that the left-hand side simplifies to

$$(2m - 1) - (2m + 1)x + x^m + x^{m+1},$$

so that the inequality becomes

$$2m - 1 \leq (2m + 1)x.$$

Since $x > 1$, the last inequality holds, and the proof is complete. $\qquad\square$

**Lemma A.4.** *Suppose $\eta_m = \eta < \frac{1}{1-\hat\rho}$, and Assumption 3 hold, then for any $m \geq 0$,*

$$\mathbb{E}\left[\|v_m - \overline{v}_m\|^2\right] \leq \frac{1 + \alpha_\eta^2}{\left(1 + \sqrt{\alpha_\eta}\right)^2} \sigma_\eta + O\left(\max\{\rho_\eta, \alpha_\eta\}^m\right),$$

*where*

$$\sigma_\eta = \frac{2\eta L_f^2 \hat\rho^2}{(1 - \hat\rho)^2(2 - \eta(1 - \rho))(1 - \rho)}, \quad \rho_\eta = (1 - \eta + \eta\hat\rho)^2, \quad \alpha_\eta = (1 - \eta + \eta\alpha)^2.$$

*Proof.* Throughout this proof, we assume $\mathbb{E}\left[\|\hat{A}_j - A\|^2\right] \neq 0$, because if $\|\hat{A}_j - A\|^2 = 0$ almost surely for all $j$, then there is no error to bound and we directly obtain $\mathbb{E}\left[\|v_m - \overline{v}_m\|^2\right] = 0$, which trivially satisfies the desired inequality.

We begin by introducing the following notation:

$$X_{k,j} = P^{k-1-j}, \quad \hat{X}_{k,j} = \prod_{i=j}^{k-1} \hat{P}_i, \quad \hat{Y}_k = \eta \sum_{j=0}^{k-1} \left(\prod_{i=j+1}^{k-1} \hat{P}_i\right) = \eta \sum_{j=0}^{k-1} \hat{X}_{k,j+1}.$$

From Lemma A.2, the difference $v_m - \overline{v}_m$ can be expressed in terms of the deviations $\hat{A}_j - A$. Specifically,

$$v_m - \overline{v}_m = \eta \sum_{k=0}^{m-1} r^{m-1-k} \sum_{j=0}^{k-1} \eta \left\{\alpha X_{k,j} \left(\hat{A}_j - A\right) \hat{X}_{j,0} + \overline{\alpha} \left(X_{k,j}(\hat{A}_j - A)(\hat{X}_{j,0} + \hat{Y}_j)\right)\right\} c$$

$$= \eta^2 \sum_{k=0}^{m-1} r^{m-1-k} \sum_{j=0}^{k-1} X_{k,j} \left(\hat{A}_j - A\right) \left(\hat{X}_{j,0} + \overline{\alpha}\hat{Y}_j\right) c.$$

Letting

$$Z_k = r^{m-k-1} \sum_{j=0}^{k-1} X_{k,j}(\hat{A}_j - A) \left(\hat{X}_{j,0} + \overline{\alpha}\hat{Y}_j\right) c,$$

we then write

$$v_m - \overline{v}_m = \eta^2 \sum_{k=0}^{m-1} Z_k.$$

(Bounding $\mathbb{E}[\|v_m - \overline{v}_m\|^2]$ using the martingale sequence) $Z_k$ is the martingale sequence with respect to its input $\mathcal{F}_{k-1} = (\hat{A}_0, \ldots, \hat{A}_{k-1})$ because from $\mathbb{E}[\hat{A}_{k-1} - A] = 0$, we have

$$\mathbb{E}[Z_k \mid \mathcal{F}_{k-2}, \ldots \mathcal{F}_0] = \mathbb{E}[Z_k \mid \mathcal{F}_{k-2}]$$

$$= r^{m-k-1} \left\{ \sum_{j=0}^{k-2} X_{k,j}(\hat{A}_j - A)\left(\hat{X}_{j,0} + \overline{\alpha}\hat{Y}_j\right) + X_{k,k-1}\mathbb{E}\left[\hat{A}_{k-1} - A\right]\left(\hat{X}_{k-1,0} + \overline{\alpha}\hat{Y}_{k-1}\right) \right\}$$

$$= Z_{k-1},$$

Using this property, we compute

$$\mathbb{E}\left[\left\|\sum_{k=0}^{m-1} Z_k\right\|^2\right] = \sum_{k=0}^{m-1} \mathbb{E}\left[\|Z_k\|^2\right] + 2\sum_{k=0}^{m-1}\sum_{j=k+1}^{m-1} \mathbb{E}\left[Z_k^\top Z_j\right]$$

$$= \sum_{k=0}^{m-1} \mathbb{E}\left[\|Z_k\|^2\right] + 2\sum_{k=0}^{m-1}\sum_{j=k+1}^{m-1} \mathbb{E}\left[Z_k^\top \underbrace{\mathbb{E}[Z_j \mid \mathcal{F}_k]}_{=Z_k}\right]$$

$$= \sum_{k=0}^{m-1} \mathbb{E}\left[\|Z_k\|^2\right] + 2\sum_{k=0}^{m-1}(m-1-k)\mathbb{E}\left[\|Z_k\|^2\right]$$

$$= \sum_{k=0}^{m-1}(2(m-k)-1)\mathbb{E}\left[\|Z_k\|^2\right]$$

Hence,

$$\mathbb{E}\left[\|v_m - \overline{v}_m\|^2\right] \le \eta^4 \mathbb{E}\left[\left\|\sum_{k=0}^{m-1} Z_k\right\|^2\right]$$

$$= \eta^4 \sum_{k=0}^{m-1}(2(m-k)-1)r^{2(m-k-1)}\mathbb{E}\left[\left\|\sum_{j=0}^{k-1} X_{k,j}(\hat{A}_j - A)\left(\hat{X}_{j,0} + \overline{\alpha}\hat{Y}_j\right)c\right\|^2\right]$$

$$= \eta^4 \sum_{k=0}^{m-1}(2(m-k)-1)r^{2(m-k-1)}\sum_{j=0}^{k-1}\|X_{k,j}\|^2\mathbb{E}\left[\left\|(\hat{A}_j - A)(\hat{X}_{j,0} + \overline{\alpha}\hat{Y}_j)c\right\|^2\right],$$

where the last equation uses the property of martingale *difference* sequence:

$$\mathbb{E}\left[X_{k,j}(\hat{A}_j - A)\left(\hat{X}_{j,0} + \overline{\alpha}\hat{Y}_j\right) \mid \mathcal{F}_{j-2}, \ldots \mathcal{F}_0\right] = 0.$$

(Bounding the inner variance norm) By expanding the variance, for some random vector $X$, we have a common property

$$\mathbb{E}\left[\|X - \mathbb{E}[X]\|^2\right] = \mathbb{E}\left[\|X\|^2\right] - 2\mathbb{E}\left[X^\top \mathbb{E}[X]\right] + \|\mathbb{E}[X]\|^2$$

$$= \mathbb{E}\left[\|X\|^2\right] - \|\mathbb{E}[X]\|^2$$

So applying this to

$$X = (\hat{A}_j - A)(\hat{X}_{j,0} + \overline{\alpha}\hat{Y}_j)c,$$

we condition on $\mathcal{F}_{j-1}$ to get

$$\mathbb{E}\left[\|(\hat{A}_j - A)(\hat{X}_{j,0} + \overline{\alpha}\hat{Y}_j)c\|^2 \mid \mathcal{F}_{j-1}\right] = \mathbb{E}[\|\hat{A}_j(\hat{X}_{j,0} + \overline{\alpha}\hat{Y}_j)c\|^2 \mid \mathcal{F}_{j-1}] - \underbrace{\mathbb{E}\left[\|A(\hat{X}_{j,0} + \overline{\alpha}\hat{Y}_j)c\|^2 \mid \mathcal{F}_{j-1}\right]}_{=\|A(\hat{X}_{j,0} + \overline{\alpha}\hat{Y}_j)c\|^2}.$$

Bounding $\|\hat{A}_j\| \le \hat{\rho}$ from Assumption 3, we get

$$\mathbb{E}\left[\|(\hat{A}_j - A)(\hat{X}_{j,0} + \overline{\alpha}\hat{Y}_j)c\|^2\right] \le \hat{\rho}^2 \|c\|^2 \mathbb{E}\left[\|\hat{X}_{j,0} + \overline{\alpha}\hat{Y}_j\|^2\right].$$

(Handling the geometric sums) Substituting into the earlier inequality, we obtain the error bound:

$$\mathbb{E}\left[\|v_m - \overline{v}_m\|^2\right] \le \eta^4 \|c\|^2 \hat{\rho}^2 \sum_{k=0}^{m-1} (2(m-k) - 1) r^{2(m-k-1)} \sum_{j=0}^{k-1} \|X_{k,j}\|^2 \mathbb{E}\left[\|\hat{X}_{j,0} + \overline{\alpha}\hat{Y}_j\|^2\right]. \qquad (37)$$

To simplify the notations for the bound in the summation, let

$$\hat{R} = (1 - \eta) + \eta\hat{\rho}, \quad R = (1 - \eta) + \eta\rho, \quad r = (1 - \eta) + \eta\alpha.$$

Then bound each of $\|X_{k,j}\|$, $\|\hat{X}_{j,0}\|$, and $\|\hat{Y}_j\|$ is obtained by geometric terms in $R$ or $\hat{R}$:

$$\|X_{k,j}\| \le R^{k-j-1}, \quad \|\hat{X}_{j,0}\| \le \hat{R}^j, \quad \|\hat{Y}_j\| \le \eta \sum_{s=0}^{j-1} \hat{R}^{j-s-1} = \frac{\eta}{1 - \hat{R}}\left(1 - \hat{R}^j\right).$$

It follows that

$$\mathbb{E}\left[\|\hat{X}_{j,0} + \overline{\alpha}\hat{Y}_j\|^2\right] \le \left(\left(1 - \overline{\alpha}\frac{\eta}{1 - \hat{R}}\right)\hat{R}^j + \overline{\alpha}\frac{\eta}{1 - \hat{R}}\right)^2$$

$$\le 2\left(1 - \overline{\alpha}\frac{\eta}{1 - \hat{R}}\right)^2 \hat{R}^{2j} + 2\left(\overline{\alpha}\frac{\eta}{1 - \hat{R}}\right)^2$$

where the last inequality uses $\|a + b\|^2 \le 2\|a\|^2 + 2\|b\|^2$. Let's define the following symbols to simplify our expressions:

$$R_p = R^2, \quad \hat{R}_p = \hat{R}^2, \quad r_p = r^2, \quad \omega = \frac{R_p}{r_p},$$

$$\kappa_1 = 2\left(1 - \overline{\alpha}\frac{\eta}{1 - \hat{R}}\right)^2, \quad \kappa_2 = 2\left(\overline{\alpha}\frac{\eta}{1 - \hat{R}}\right)^2, \quad \pi_1 = \frac{\hat{R}_p}{R_p}, \quad \pi_2 = R_p^{-1},$$

Using these definitions, we can continue our derivation:

$$\|X_{k,j}\|^2 \mathbb{E}\left[\|\hat{X}_{j,0} + \overline{\alpha}\hat{Y}_j\|^2\right] \le R^{2(k-j-1)}\left[2\left(1 - \overline{\alpha}\frac{\eta}{1 - \hat{R}}\right)^2 \hat{R}^{2j} + 2\left(\overline{\alpha}\frac{\eta}{1 - \hat{R}}\right)^2\right]$$

$$= R_p^{k-j-1}\left[\kappa_1 \hat{R}_p^j + \kappa_2\right]$$

$$= R_p^{k-j-1}\kappa_1 \hat{R}_p^j + R_p^{k-j-1}\kappa_2$$

$$= R_p^{k-1}\left[\kappa_1 \frac{\hat{R}_p^j}{R_p^j} + \kappa_2 \frac{1}{R_p^j}\right]$$

$$= R_p^{k-1}\left[\kappa_1 \pi_1^j + \kappa_2 \pi_2^j\right]$$

Using the above derivation, (37) is expressed as

$$\mathbb{E}\left[\|v_m - \overline{v}_m\|^2\right] \leq \eta^4 \|c\|^2 \hat{\rho}^2 \frac{1}{R_p} r_p^{m-1} \sum_{k=0}^{m-1} (2(m-k) - 1)\omega^k \left[\kappa_1 \sum_{j=0}^{k-1} \pi_1^j + \kappa_2 \sum_{j=0}^{k-1} \pi_2^j\right]. \tag{38}$$

We then split this into a decaying part and a non-decaying part.

First, consider the first term of the decaying factor in (38). Recalling $\pi_1 > 1$ implied by $\mathbb{E}\left[\|\hat{A}_j - A\|^2\right] \neq 0$ and Lemma A.3, we have

$$
\begin{aligned}
\kappa_1 \frac{1}{R_p} r_p^{m-1} \sum_{k=0}^{m-1} (2(m-k) - 1)\omega^k \sum_{j=0}^{k-1} \pi_1^j &= \kappa_1 \frac{1}{R_p} \frac{1}{\pi_1 - 1} r_p^{m-1} \sum_{k=0}^{m-1} (2(m-k) - 1)\omega^k \left(\pi_1^k - 1\right) \\
&\leq \kappa_1 \frac{1}{R_p} \frac{1}{\pi_1 - 1} r_p^{m-1} \left(\frac{(\omega\pi_1)^{m+1}}{(\omega\pi_1 - 1)^2}\right) \\
&= \kappa_1 \frac{1}{\frac{\hat{R}_p}{R_p} - 1} \frac{1}{R_p} r_p^{m-1} \frac{\left(\frac{\hat{R}_p}{r_p}\right)^{m+1}}{\left(\frac{\hat{R}_p}{r_p} - 1\right)^2} \\
&= \kappa_1 \frac{1}{r_p^2 \left(\hat{R}_p - R_p\right)} \frac{\hat{R}_p^{m+1} - r_p^{m+1}}{\left(\frac{\hat{R}_p}{r_p} - 1\right)^2} = O\left(\max\{\hat{R}_p, \hat{r}_p\}^m\right). \tag{39}
\end{aligned}
$$

Now consider the second term of (38), which is the non-decaying part. Similarly, we use Lemma A.3 and $\pi_2 > 1$ from $\mathbb{E}\left[\|\hat{A}_j - A\|^2\right] \neq 0$, having

$$
\begin{aligned}
\kappa_2 \frac{1}{R_p} r_p^{m-1} \sum_{k=0}^{m-1} (2(m-k) - 1)\omega^k \sum_{j=0}^{k-1} \pi_2^j &= \kappa_2 \frac{1}{R_p} \frac{1}{\pi_2 - 1} r_p^{m-1} \sum_{k=0}^{m-1} (2(m-k) - 1)\omega^k \left(\pi_2^k - 1\right) \\
&\leq \kappa_2 \frac{1}{R_p} \frac{1}{\pi_2 - 1} r_p^{m-1} \frac{(\omega\pi_2)^m (\omega\pi_2 + 1)}{(\omega\pi_2 - 1)^2} \\
&= \kappa_2 \frac{1}{1 - R_p} r_p^{m-1} \frac{\left(\frac{1}{r_p}\right)^m \left(1 + \frac{1}{r_p}\right)}{\left(\frac{1}{r_p} - 1\right)^2} \\
&= \kappa_2 \frac{1}{1 - R_p} \frac{1 + r_p}{(1 - r_p)^2}.
\end{aligned}
$$

Since the denominator can be replaced with

$$
\begin{aligned}
1 - R^2 &= 1 - ((1 - \eta) + \eta\rho)^2 = \eta (2 - \eta (1 - \rho)) (1 - \rho), \\
(1 - \hat{R})^2 &= (1 - ((1 - \eta) + \eta\hat{\rho}))^2 = \eta^2 (1 - \hat{\rho})^2, \\
(1 - r^2)^2 &= \left(1 - (1 - \eta(1 - \alpha))^2)\right)^2 = \eta^2 (2 - \eta(1 - \alpha))^2 (1 - \alpha)^2,
\end{aligned}
$$

the non-decaying part is finally expressed as

$$\kappa_2 \frac{1}{R_p} r_p^{m-1} \sum_{k=0}^{m-1} (2(m-k) - 1)\omega^k \sum_{j=0}^{k-1} \pi_2^j \leq 2 \frac{1 + (1 - \eta(1 - \alpha))^2}{\eta^3 \hat{\rho}^2 (1 - \hat{\rho})^2 (2 - \eta(1 - \rho)) (1 - \rho) (2 - \eta(1 - \alpha))^2}. \tag{40}$$

By injecting the decaying term (39) and non-decaying term (40) into (38), we present the final result as:

$$\mathbb{E}\left[\|v_m - \overline{v}_m\|^2\right] \leq 2L_f^2\hat{\rho}^2 \frac{\eta(1 + (1 - \eta(1-\alpha))^2)}{(1-\hat{\rho})^2\,(2 - \eta(1-\rho))\,(1-\rho)\,(2 - \eta(1-\alpha))^2} + O\left(\max\{\rho_\eta, \alpha_\eta\}^m\right)$$

$$= \frac{1 + (1 - \eta(1-\alpha))^2}{(2 - \eta(1-\alpha))^2}\sigma_\eta + O\left(\max\{\rho_\eta, \alpha_\eta\}^m\right)$$

$$= \frac{1 + \alpha_\eta^2}{\left(1 + \sqrt{\alpha_\eta}\right)^2}\sigma_\eta + O\left(\max\{\rho_\eta, \alpha_\eta\}^m\right).$$

$\square$

### A.6  Convergence Analysis of Deterministic MixedFP-KM

**Lemma A.5.** *Suppose $\eta_m = \eta < \frac{1}{1-\hat{\rho}}$ and Assumption 3 hold, then for any $m \geq 0$,*

$$\|\overline{v}_m - \mathrm{d}_x f\|^2 = O\left(\max\{\overline{\rho}_\eta, \alpha_\eta\}^m\right).$$

*where*

$$\overline{\rho}_\eta = (1 - \eta + \eta\rho)^2, \quad \alpha_\eta = (1 - \eta + \eta\alpha)^2.$$

*Proof.* From Lemma A.1, we have

$$\begin{aligned}
\overline{v}_m &= \eta \sum_{k=0}^{m-1} r^{m-k-1}\left(\alpha\overline{u}_k + \overline{\alpha}\overline{w}_k\right) \\
&= \eta \sum_{k=0}^{m-1} r^{m-k-1}\left(\alpha P^k c + \overline{\alpha}\left(P^k c + \eta \sum_{s=0}^{k-1} P^{k-s-1} c\right)\right) \\
&= \eta \sum_{k=0}^{m-1} r^{m-k-1}\left(P^k c + \overline{\alpha}\eta \sum_{s=0}^{k-1} P^{k-s-1} c\right).
\end{aligned} \tag{41}$$

Using the identity for the matrix inverse $\sum_{i=0}^{n-1} X^i = (I - X^n)(I - X)^{-1}$ and the fact that

$$\begin{aligned}
\eta(I - P)^{-1} c &= \eta\left(I - (I - \eta(I - A))\right)^{-1} c \\
&= \eta\left(\eta(I - A)\right)^{-1} c = \mathrm{d}_x f,
\end{aligned}$$

the term inside the sum in (41) can be rewritten as

$$\begin{aligned}
P^k c + \overline{\alpha}\eta \sum_{s=0}^{k-1} P^{k-s-1} c &= P^k c + \overline{\alpha}\eta\left((I - P^k)(I - P)^{-1}\right) c \\
&= P^k\left(I - \overline{\alpha}(I - A)^{-1}\right) c + \overline{\alpha}\mathrm{d}_x f.
\end{aligned}$$

By letting $D = I - \overline{\alpha}(I - A)^{-1}$ and substituting them into (41), we obtain

$$\begin{aligned}
\overline{v}_m - \mathrm{d}_x f &= \eta \sum_{k=0}^{m-1} r^{m-k-1}\left(P^k D c + \overline{\alpha}\mathrm{d}_x f\right) - \mathrm{d}_x f \\
&= \eta \sum_{k=0}^{m-1} r^{m-k-1} P^k D c + \underbrace{\frac{\overline{\alpha}\eta}{1 - r}}_{=1}(1 - r^m)\,\mathrm{d}_x f - \mathrm{d}_x f \\
&= \eta \sum_{k=0}^{m-1} r^{m-k-1} P^k D - r^m \mathrm{d}_x f.
\end{aligned}$$

Using $\|a + b\|^2 \leq 2 \|a\|^2 + 2 \|b\|^2$, we have

$$
\begin{aligned}
\|\overline{v}_m - \mathrm{d}_x f\|^2 &\leq 2\eta^2 \|D\|^2 \left( \sum_{k=0}^{m-1} r^{m-k-1} R^k \right)^2 \|c\|^2 + 2 \left( \frac{r^m}{1-\rho} \right)^2 \|c\|^2 \\
&= 2\eta^2 \|D\|^2 \left( \frac{R^m - r^m}{R - r} \right)^2 \|c\|^2 + 2 \left( \frac{r^m}{1-\rho} \right)^2 \|c\|^2 \\
&= O\left( \max\{R, r\}^{2m} \right) = O\left( \max\{\overline{\rho}_\eta, \alpha_\eta\}^m \right).
\end{aligned}
$$

$\square$

### A.7 Main Convergence Theorems

**Theorem 4.5** (MixedFP-KM). *Let $\alpha \in [0, 1)$. When $\eta_m = \eta < \frac{1}{1-\hat{\rho}}$ and Assumptions 1 to 3 hold, then for any $m \geq 0$,*

$$
\mathbb{E}\left[ \|v_m - \mathrm{d}_x f\|^2 \right] \leq \frac{1 + \alpha_\eta}{\left( 1 + \sqrt{\alpha_\eta} \right)^2} \sigma_\eta + O\left( \max\{\rho_\eta, \alpha_\eta\}^m \right),
$$

*where*

$$
\sigma_\eta = \frac{2\eta L_f^2 \hat{\rho}^2}{(1 - \hat{\rho})^2 (2 - \eta(1 - \rho))(1 - \rho)}, \quad \rho_\eta = (1 - \eta + \eta\hat{\rho})^2, \quad \alpha_\eta = (1 - \eta + \eta\alpha)^2.
$$

*Proof.* We divide the difference into

$$
v_m - \mathrm{d}_x f = (v_m - \overline{v}_m) + (\overline{v}_m - \mathrm{d}_x f),
$$

where $\overline{v}_m$ is the deterministic counterpart. Then

$$
\mathbb{E}\left[ \|v_m - \mathrm{d}_x f\|^2 \right] \leq \mathbb{E}\left[ \|v_m - \overline{v}_m\|^2 \right] + \mathbb{E}\left[ \|\overline{v}_m - \mathrm{d}_x f\|^2 \right],
$$

where the cross term vanishes because $\mathbb{E}[v_m - \overline{v}_m] = 0$. Applying Lemma A.4 (for $\|v_m - \overline{v}_m\|$) and Lemma A.5 (for $\|\overline{v}_m - \mathrm{d}_x f\|$) proves the claim.

$\square$

We obtain Theorem 4.1 as the special case of Theorem 4.5 where $\eta = 1$ with additional consideration for $\alpha = 1$.

**Theorem 4.1** (MixedFP). *Suppose Assumptions 1 to 3 hold and $\alpha \in [0, 1]$, then for any $m \geq 0$,*

$$
\mathbb{E}\left[ \|\hat{v}_m - \mathrm{d}_x f\|^2 \right] \leq
\begin{cases}
\sigma_1 + O\left( \rho_1^m \right) & \text{if } \alpha \in \{0, 1\}, \\
\dfrac{1 + \alpha_1}{(1 + \sqrt{\alpha_1})^2} \sigma_1 + O\left( \max\{\rho_1, \alpha_1\}^m \right) & \text{otherwise},
\end{cases}
$$

*where*

$$
\sigma_1 = \frac{2 L_f^2 \hat{\rho}^2}{(1 - \hat{\rho})^2 (1 - \rho^2)}, \quad \rho_1 = \hat{\rho}^2, \quad \alpha_1 = \alpha^2.
$$

*Proof.* For $\alpha \in [0, 1)$, this result is the special case of Theorem 4.5 where $\eta = 1$, which gives $\alpha_\eta = \alpha^2 = \alpha_1$, $\rho_\eta = \hat{\rho}^2 = \rho_1$, and $\sigma_\eta = \frac{2 L_f^2 \hat{\rho}^2}{(1 - \hat{\rho})^2 (1 - \rho^2)} = \sigma_1$. When $\alpha = 1$, $\hat{v}_m$ is equivalent to $\hat{u}_{m-1}$ whose error is given by Corollary B.1.

$\square$

# B Theoretical Analysis of StocFP with the Stochastic KM Iteration

## B.1 Convergence of StocFP with the Stochastic KM Iteration

**Theorem 4.6** (Stochastic KM iteration of StocFP). *Suppose $\eta_m = \eta < \frac{1}{1-\hat{\rho}}$ and Assumptions 1 to 3 hold. Then for any $m \geq 0$,*

$$\mathbb{E}\left[\|w_m - \mathrm{d}_x f\|^2\right] \leq \sigma_\eta + O\left(\rho_\eta^m\right),$$

*where $\sigma_\eta$ and $\rho_\eta$ are as defined in Theorem 4.5.*

*Proof.* The proof is divided into two parts: the variance bound (that is, the stochastic error between $w_m$ and $\overline{w}_m$), and the deterministic error bound (i.e., the bias $\|\overline{w}_m - \mathrm{d}_x f\|^2$). We then combine these two parts.

From (29), the error in the $w$–sequence can be written as

$$w_m - \overline{w}_m = \eta \sum_{j=0}^{m-1} P^{m-1-j}(\hat{A}_j - A)\left\{\left(\prod_{i=0}^{j-1}\hat{P}_i\right)c + \eta\sum_{k=0}^{j-1}\left(\prod_{i=k+1}^{j-1}\hat{P}_i\right)c\right\}.$$

Recall

$$X_{m,j} := P^{m-1-j}, \quad \hat{X}_{j,0} := \prod_{i=0}^{j-1}\hat{P}_i, \quad \hat{Y}_j := \eta\sum_{k=0}^{j-1}\prod_{i=k+1}^{j-1}\hat{P}_i.$$

Then,

$$w_m - \overline{w}_m = \eta \sum_{j=0}^{m-1} X_{m,j}(\hat{A}_j - A)\left(\hat{X}_{j,0} + \hat{Y}_j\right)c.$$

Because $\mathbb{E}[\hat{A}_j] = A$ from Assumption 3, the sequence

$$W_j := X_{m,j}(\hat{A}_j - A)\left(\hat{X}_{j,0} + \hat{Y}_j\right)c$$

forms a martingale–difference sequence with respect to the natural filtration $\mathcal{F}_j = \sigma(\hat{A}_0, \ldots, \hat{A}_j)$. Hence, by orthogonality of martingale differences and the Cauchy–Schwarz inequality,

$$\mathbb{E}\left[\|w_m - \overline{w}_m\|^2\right] = \mathbb{E}\left[\left\|\eta\sum_{j=0}^{m-1}W_j\right\|^2\right]$$

$$\leq \eta^2\mathbb{E}\left[\left\|\sum_{j=0}^{m-1}W_j\right\|^2\right]$$

$$= \eta^2\sum_{j=0}^{m-1}\mathbb{E}\left[\|W_j\|^2\right].$$

By expanding the term in the summation, we have

$$\mathbb{E}\left[\|W_j\|^2\right] = \mathbb{E}\left[\|X_{m,j}(\hat{A}_j - A)(\hat{X}_{j,0} + \hat{Y}_j)c\|^2\right]$$

$$= \|X_{m,j}\|^2\mathbb{E}\left[\|(\hat{A}_j - A)(\hat{X}_{j,0} + \hat{Y}_j)c\|^2\right]$$

Conditioning on $\mathcal{F}_{j-1}$, we get

$$\mathbb{E}\left[\|(\hat{A}_j - A)(\hat{X}_{j,0} + \hat{Y}_j)c\|^2 \mid \mathcal{F}_{j-1}\right] = \mathbb{E}\left[\|\hat{A}_j(\hat{X}_{j,0} + \hat{Y}_j)c\|^2 \mid \mathcal{F}_{j-1}\right] - \|A(\hat{X}_{j,0} + \hat{Y}_j)c\|^2$$

where we used the $\mathbb{E}\left[\|X - \mathbb{E}[X]\|^2\right] = \mathbb{E}\left[\|X\|^2\right] - \|\mathbb{E}[X]\|^2$. From Assumption 3, we have $\|\hat{A}_j\| \le \hat{\rho}$, which gives us

$$\mathbb{E}\left[\|(\hat{A}_j - A)(\hat{X}_{j,0} + \hat{Y}_j)c\|^2\right] \le \hat{\rho}^2 \|c\|^2 \mathbb{E}\left[\|\hat{X}_{j,0} + \hat{Y}_j\|^2\right]$$

Therefore,

$$\mathbb{E}\left[\|W_j\|^2\right] \le \|X_{m,j}\|^2 \hat{\rho}^2 \|c\|^2 \mathbb{E}\left[\|\hat{X}_{j,0} + \hat{Y}_j\|^2\right] \tag{42}$$

Furthermore, since the norm of $P = I - \eta(I - A)$ is bounded by $\|P\| \le R$ with $R = (1-\eta) + \eta\rho = 1 - \eta(1-\rho)$. Thus,

$$\|X_{m,j}\| \le R^{m-1-j}$$

Similarly, the product of the stochastic matrices satisfies

$$\|\hat{X}_{j,0}\| \le \hat{R}^j, \quad \|\hat{Y}_j\| \le \eta \sum_{k=0}^{j-1} \hat{R}^{j-k-1} = \frac{\eta}{1-\hat{R}}\left(1 - \hat{R}^j\right)$$

with $\hat{R} = (1-\eta) + \eta\hat{\rho} = 1 - \eta(1-\hat{\rho})$. Therefore,

$$\|\hat{X}_{j,0} + \hat{Y}_j\|^2 \le \left(\hat{R}^j + \frac{\eta}{1-\hat{R}}\left(1 - \hat{R}^j\right)\right)^2 .$$
$$\le 2\left(1 - \frac{\eta}{1-\hat{R}}\right)^2 \hat{R}^{2j} + 2\left(\frac{\eta}{1-\hat{R}}\right)^2$$

Substituting them into (42) gives

$$\mathbb{E}\left[\|w_m - \overline{w}_m\|^2\right] \le \eta^2 \|c\|^2 \hat{\rho}^2 \sum_{j=0}^{m-1} R^{2(m-1-j)}\left(2\left(1 - \frac{\eta}{1-\hat{R}}\right)^2 \hat{R}^{2j} + 2\left(\frac{\eta}{1-\hat{R}}\right)^2\right). \tag{43}$$

By letting

$$\kappa_3 = 2\left(1 - \frac{\eta}{1-R}\right)^2, \quad \kappa_4 = 2\left(\frac{\eta}{1-R}\right)^2, \quad R_p = R^2, \quad \hat{R}_p = \hat{R}^2, \quad \pi_1 = \frac{\hat{R}_p}{R_p}, \quad \pi_2 = R_p^{-1},$$

We simplify the notation of as (43) as

$$\mathbb{E}\left[\|w_m - \overline{w}_m\|^2\right] \le \eta^2 \|c\|^2 \hat{\rho}^2 R_p^{m-1} \sum_{k=0}^{m-1}\left(\kappa_3 \pi_1^k + \kappa_4 \pi_2^k\right).$$

Since $R = 1 - \eta(1-\rho) < \hat{R} = 1 - \eta(1-\hat{\rho}) < 1$ from $\hat{\rho} > \rho$ and $0 < \eta < \frac{1}{1-\hat{\rho}}$, we have

$$R_p^{m-1} \sum_{k=0}^{m-1}\left(\kappa_3 \pi_1^k + \kappa_4 \pi_2^k\right) = R_p^{m-1}\left(\kappa_3 \frac{\pi_1^k - 1}{\pi_1 - 1} + \kappa_4 \frac{\pi_2^k - 1}{\pi_2 - 1}\right)$$
$$\le \left(\kappa_3 \frac{\hat{R}_p^m}{\hat{R}_p - R_p} + \kappa_4 \frac{1}{1 - R_p}\right)$$
$$= \kappa_4 \frac{1}{1 - R_p} + O\left(\hat{R}_p^m\right)$$
$$= \frac{2}{\eta(1-\hat{\rho})^2(2 - \eta(1-\rho))(1-\rho)} + O\left(\hat{R}_p^m\right)$$

It follows that

$$\mathbb{E}\left[\|w_m - \overline{w}_m\|^2\right] \leq 2L_f^2 \hat{\rho}^2 \frac{\eta}{(1-\hat{\rho})^2 \left(2 - \eta(1-\rho)\right)(1-\rho)} + O\left((1 - \eta(1-\hat{\rho}))^{2m}\right).$$

We now derive in full the bound for the deterministic error $\|\overline{w}_m - \mathrm{d}_x f\|^2$. The deterministic iteration is given by

$$\overline{w}_0 = c, \quad \overline{w}_{m+1} = P\overline{w}_m + \eta c, \quad \text{with} \quad P = I - \eta(I - A).$$

Unrolling this recursion yields

$$\overline{w}_m = P^m c + \eta \sum_{k=0}^{m-1} P^{m-1-k} c.$$

Changing the index (letting $j = m - 1 - k$) we have

$$\overline{w}_m = P^m c + \eta \sum_{j=0}^{m-1} P^j = P^m c + \eta \left(I - P^m\right)\left(I - P\right)^{-1} c.$$

Subtracting $\mathrm{d}_x f$ from $\overline{w}_m$ yields

$$\begin{aligned}
\overline{w}_m - \mathrm{d}_x f &= \left[P^m c + \eta\left(I - P^m\right)\left(I - P\right)^{-1} c\right] - \eta(I - P)^{-1} c \\
&= P^m c + \eta\left[(I - P^m) - I\right](I - P)^{-1} c \\
&= P^m c - \eta P^m (I - P)^{-1} c \\
&= P^m \left[c - \eta(I - P)^{-1} c\right].
\end{aligned}$$

Since $\mathrm{d}_x f = \eta(I - P)^{-1} c$, we rewrite this as

$$\overline{w}_m - \mathrm{d}_x f = P^m \left(c - \mathrm{d}_x f\right).$$

As $\|P\| \leq 1 - \eta(1-\rho) =: R$ holds under Assumption 3, we deduce that

$$\|\overline{w}_m - \mathrm{d}_x f\|^2 = O\left((1 - \eta(1-\rho))^{2m}\right).$$

We now decompose the overall error as

$$w_m - \mathrm{d}_x f = (w_m - \overline{w}_m) + (\overline{w}_m - \mathrm{d}_x f).$$

Since $\mathbb{E}\left[w_m - \overline{w}_m\right] = 0$, we have

$$\mathbb{E}\left[\|w_m - \mathrm{d}_x f\|^2\right] \leq \mathbb{E}\left[\|w_m - \overline{w}_m\|^2\right] + \|\overline{w}_m - \mathrm{d}_x f\|^2.$$

Inserting the bounds derived above yields

$$\mathbb{E}\left[\|w_m - \mathrm{d}_x f\|^2\right] \leq 2L_f^2 \hat{\rho}^2 \frac{\eta}{(1-\hat{\rho})^2 \left(2 - \eta(1-\rho)\right)(1-\rho)} + O\left((1 - \eta(1-\hat{\rho}))^{2m}\right).$$

$\square$

We obtain the error bounds for StocFP (4) StocRB (5) as the special case of Theorem 4.6 where $\eta = 1$.

**Corollary B.1** (StocFP and StocRB). *Suppose Assumptions 1 to 3 hold. Then for any $m \geq 0$,*

$$\mathbb{E}\left[\|\hat{w}_m - \mathrm{d}_x f\|^2\right] = \mathbb{E}\left[\|\hat{y}_m - \mathrm{d}_x f\|^2\right] \leq \sigma_1 + O\left(\rho_1^m\right),$$

*where $\sigma_1$ and $\rho_1$ are as defined in Theorem 4.1.*

## B.2 Comparison with the Existing Result

Theorem 4.6 for StocFP with the stochastic KM iteration (StocFP-KM) slightly differs from a bound derived from results in the original paper (Grazzi et al., 2021).

Grazzi et al. (2021, Theorem 5.1) presents a non-asymptotic error bound for StocFP-KM with a decreasing stepsize, while the bound for a fixed stepsize is partially provided in their Theorem 4.1, which is a non-asymptotic version of our Theorem 4.2. By combining their Theorem 4.1 and their proof of Theorem 5.1, we can derive the following result for StocFP-KM with a constant stepsize.

**Theorem B.2** (Fixed stepsize and asymptotic version of Grazzi et al. (2021, Theorem 5.1)). *Let* $\sigma_2 = \frac{2(\hat{\rho}^2 + \rho^2)}{(1-\rho)^2}$ *and suppose* $0 < \eta \leq \frac{1}{1+\sigma_2}$ *and Assumptions 1 to 3 hold. Then for any* $m \geq 0$,

$$\mathbb{E}\left[\|w_m - \mathrm{d}_x f\|^2\right] \leq 2L_f^2 \hat{\rho}^2 \frac{\eta}{(1-\rho^2)(1-\rho)^2} + O\left((1 - \eta(1 - \rho^2))^m\right)$$

Although both Theorem 4.6 and Theorem B.2 evaluate the same method, the derivations differ. We chose to evaluate the error of the existing method using an approach similar to our own analysis in Theorem 4.5 in order to ensure a fair comparison between our proposed method and that of Grazzi et al. (2021). Consequently, our Theorem 4.6 does not unfairly penalize the method in Grazzi et al. (2021); in fact, there are cases where Theorem 4.6 even yields tighter bounds than Theorem B.2. One example is obtained by choosing

$$\rho = 0.1, \quad \hat{\rho} = 0.2, \quad \eta = 10^{-2}.$$

These values satisfy the side conditions $0 < \rho < 1$, $\rho < \hat{\rho}$, and $\eta < \frac{1}{1-\hat{\rho}}$. By dividing both errors by $2\eta L_f^2 \hat{\rho}^2$, we have

$$\underbrace{\frac{1}{(1-\hat{\rho})^2(2-\eta(1-\rho))(1-\rho)}}_{\approx 0.872} + O\left(\left(\underbrace{(1-\eta(1-\hat{\rho}))^2}_{=0.984}\right)^m\right) < \underbrace{\frac{1}{(1-\rho^2)(1-\rho)^2}}_{\approx 1.247} + O\left(\left(\underbrace{1-\eta(1-\rho^2)}_{=0.990}\right)^m\right)$$

This is an example where both the non-decaying term and the decaying term of our Theorem 4.6 are superior to Theorem B.2 obtained from the existing results.

## C  Impact of Approximate Inner Solutions on Estimation Accuracy

In the previous sections, we analyzed the convergence properties of MixedFP-KM and StocFP-KM under the assumption that the inner problem is solved exactly. However, in practical applications, we often work with approximate solutions of the inner problem. This section investigates how the accuracy of hypergradient estimation is affected when using inexact inner solutions, providing theoretical guarantees for the error bounds in such scenarios.

In short, the errors induced by inexact inner solves are consistent between the mixed and stochastic estimators, and the variance term, which MixedFP–KM is expressly designed to curb, remains unchanged.

We suppose that Jacobians and gradients are evaluated at an approximate inner solution $\tilde{x} \in \mathbb{R}^{d_x}$, writing $\delta x := \tilde{x} - x(\lambda)$, introducing smoothness of stochastic Jacobians and gradients.

**Assumption 4** (Smoothness with respect to the inner parameter). *There exist finite constants* $L_A, L_c > 0$ *such that for every* $\lambda \in \mathbb{R}^{d_\lambda}$, $x, x' \in \mathbb{R}^{d_x}$, *and* $\xi \in \mathcal{X}$,

$$\|\partial_x \hat{\varphi}(x, \lambda; \xi) - \partial_x \hat{\varphi}(x', \lambda; \xi)\| \leq L_A \|x - x'\|, \quad \|\partial_x f(x, \lambda) - \partial_x f(x', \lambda)\| \leq L_c \|x - x'\|.$$

## C.1  MixedFP-KM with Inexact Inner Solution

MixedFP-KM runs with Jacobians and gradients evaluated at $\tilde{x}$, producing the stochastic sequence

$$\tilde{z}_m = (\tilde{v}_m, \tilde{w}_m, \tilde{u}_m), \quad m = 0, 1, \ldots$$

and its deterministic counterpart (same recursion, but replacing each $\partial_x \hat{\varphi}(\tilde{x}, \lambda; \xi_m)$ by its expectation) is denoted $\bar{\tilde{z}}_m = (\bar{\tilde{v}}_m, \bar{\tilde{w}}_m, \bar{\tilde{u}}_m)$.

We also use the following standard resolvent identity.

**Lemma C.1** (Resolvent identity). *Let $A, \tilde{A} \in \mathbb{R}^{d_x \times d_x}$ be such that $I - A$ and $I - \tilde{A}$ are invertible. Then*

$$(I - \tilde{A})^{-1} - (I - A)^{-1} = (I - \tilde{A})^{-1}(\tilde{A} - A)(I - A)^{-1}. \tag{44}$$

*Proof.* Starting from the right-hand side,

$$(I - \tilde{A})^{-1}(\tilde{A} - A)(I - A)^{-1} = (I - \tilde{A})^{-1}\big[(I - A) - (I - \tilde{A})\big](I - A)^{-1},$$

because $\tilde{A} - A = (I - A) - (I - \tilde{A})$. Distributing the product,

$$(I - \tilde{A})^{-1}(I - A)(I - A)^{-1} - (I - \tilde{A})^{-1}(I - \tilde{A})(I - A)^{-1} = (I - \tilde{A})^{-1} - (I - A)^{-1}.$$

This completes the proof. $\qquad\square$

**Theorem C.2** (MixedFP-KM with inexact inner solution). *Let $\delta x = \tilde{x} - x(\lambda)$. Suppose Assumptions 1 to 4 hold and that MixedFP–KM is run with Jacobians and gradients evaluated at $\tilde{x}$, generating $\tilde{v}_m$. Assume also $0 < \eta < \frac{1}{1-\hat{\rho}}$. Then for all $m \geq 0$,*

$$\mathbb{E}\big[\|\tilde{v}_m - \mathrm{d}_x f\|^2\big] \leq \frac{1 + \alpha_\eta}{(1 + \sqrt{\alpha_\eta})^2}\sigma_\eta + K\|\delta x\|^2 + O\big(\max\{\rho_\eta, \alpha_\eta\}^m\big),$$

*where $\alpha_\eta, \rho_\eta, \sigma_\eta$ are from Theorem 4.5 and*

$$K = 2\left(\frac{L_c^2}{(1-\rho)^2} + \frac{\|c\|^2 L_A^2}{(1-\rho)^4}\right).$$

*Proof.* Let $\tilde{A} = \partial_x \varphi(\tilde{x}, \lambda)$, $\tilde{c} = \partial_x f(\tilde{x}, \lambda)$. We decompose the total difference:

$$\tilde{v}_m - \mathrm{d}_x f = (\tilde{v}_m - \bar{\tilde{v}}_m) + (\bar{\tilde{v}}_m - \bar{\tilde{v}}_\infty) + (\bar{\tilde{v}}_\infty - \mathrm{d}_x f).$$

Here, $\bar{\tilde{v}}_m$ is the deterministic sequence obtained by replacing each $\partial_x \hat{\varphi}(\tilde{x}, \lambda; \xi_k)$ with $\tilde{A}$, and $\bar{\tilde{v}}_\infty = (I - \tilde{A})^{-1}\tilde{c}$. Taking norms and expectations, we use $(a + b)^2 \leq 2a^2 + 2b^2$ and note that $\mathbb{E}[\tilde{v}_m - \bar{\tilde{v}}_m] = 0$, so cross terms vanish in expectation. Thus,

$$\mathbb{E}\big[\|\tilde{v}_m - \mathrm{d}_x f\|^2\big] \leq \mathbb{E}\big[\|\tilde{v}_m - \bar{\tilde{v}}_m\|^2\big] + \|(\bar{\tilde{v}}_m - \bar{\tilde{v}}_\infty) + (\bar{\tilde{v}}_\infty - \mathrm{d}_x f)\|^2$$
$$\leq \mathbb{E}\big[\|\tilde{v}_m - \bar{\tilde{v}}_m\|^2\big] + \|\bar{\tilde{v}}_\infty - \mathrm{d}_x f\|^2 + 2\|\bar{\tilde{v}}_m - \bar{\tilde{v}}_\infty\|\|\bar{\tilde{v}}_\infty - \mathrm{d}_x f\| + \|\bar{\tilde{v}}_m - \bar{\tilde{v}}_\infty\|^2$$

First, replacing $A$ by $\tilde{A}$ and $c$ by $\tilde{c}$ in the proof of Lemma A.4 shows

$$\mathbb{E}\big[\|\tilde{v}_m - \bar{\tilde{v}}_m\|^2\big] \leq \frac{1 + \alpha_\eta}{(1 + \sqrt{\alpha_\eta})^2}\sigma_\eta + O\big(\max\{\rho_\eta, \alpha_\eta\}^m\big),$$

noting that $\|\tilde{A} - A\| \leq \hat{\rho}$. Next, $\bar{\tilde{v}}_\infty - \mathrm{d}_x f = (I - \tilde{A})^{-1}\tilde{c} - (I - A)^{-1}c$ is handled by the resolvent identity of Lemma C.1, which implies

$$(I - \tilde{A})^{-1} - (I - A)^{-1} = (I - \tilde{A})^{-1}(\tilde{A} - A)(I - A)^{-1}.$$

Hence noting $\tilde{c} = (\tilde{c} - c) + c$,

$$(I - \tilde{A})^{-1}\tilde{c} - (I - A)^{-1}c = (I - \tilde{A})^{-1}(\tilde{c} - c) + (I - \tilde{A})^{-1}(\tilde{A} - A)(I - A)^{-1}c.$$

Because $\|(I - \tilde{A})^{-1}\| \leq \frac{1}{1-\rho}$ and $\|\tilde{A} - A\| \leq L_A\|\delta x\|$, $\|\tilde{c} - c\| \leq L_c\|\delta x\|$ from Assumption 4, we get

$$\|\bar{\tilde{v}}_\infty - \mathrm{d}_x f\| \leq \frac{\|\tilde{c} - c\|}{1 - \rho} + \frac{\|\tilde{A} - A\|\|c\|}{(1 - \rho)^2} \leq \left(\frac{L_c}{1 - \rho} + \frac{\|c\|L_A}{(1 - \rho)^2}\right)\|\delta x\|.$$

Therefore its squared norm is at most

$$\|\bar{\bar{v}}_\infty - \mathrm{d}_x f\|^2 \leq 2\Big(\frac{L_c^2}{(1-\rho)^2} + \frac{\|c\|^2 L_A^2}{(1-\rho)^4}\Big)\|\delta x\|^2,$$

which is our constant term $K\|\delta x\|^2$.

Finally, by the analysis in Lemma A.5 with the single fixed matrix $\tilde{A}$ and vector $\tilde{c}$, we get

$$\|\bar{v}_m - \bar{\bar{v}}_\infty\| = O\big(\max\{R, r\}^m\big).$$

Recalling that $\|\bar{\bar{v}}_\infty - \mathrm{d}_x f\|$ is constant,

$$2\|\bar{v}_m - \bar{\bar{v}}_\infty\|\|\bar{\bar{v}}_\infty - \mathrm{d}_x f\| + \|\bar{v}_m - \bar{\bar{v}}_\infty\|^2 = O\left(\max\{\overline{\rho}_\eta, \alpha_\eta\}^m\right)$$

Collecting all three parts completes the proof. $\qquad\square$

## C.2 StocFP-KM with Inexact Inner Solution

The following result quantifies the additional error that appears when the fixed–point equation for the inner variable is solved only approximately. All assumptions stated earlier remain in force.

Let $\tilde{x}$ be an inexact inner solution and define

$$\delta x := \tilde{x} - x(\lambda), \qquad \tilde{A}_m := \partial_x \hat{\varphi}\big(\tilde{x}, \lambda; \xi_m\big), \qquad \tilde{c} := \partial_x f\big(\tilde{x}, \lambda\big).$$

For a fixed stepsize $\eta \in \big(0, \frac{1}{1-\hat{\rho}}\big)$ the stochastic Krasnosel'skiĭ–Mann (KM) version of StocFP is

$$\tilde{w}_0 = \tilde{c}, \quad \tilde{w}_{m+1} = (1-\eta)\tilde{w}_m + \eta\big(\tilde{A}_m \tilde{w}_m + \tilde{c}\big), \qquad m = 0, 1, \dots$$

**Theorem C.3** (StocFP–KM with inexact inner solution). *Let Assumptions 1 to 4 hold and let $\tilde{w}_m$ be generated as above. Then for every $m \geq 0$,*

$$\mathbb{E}\big[\|\tilde{w}_m - \mathrm{d}_x f\|^2\big] \leq \sigma_\eta + K\|\delta x\|^2 + O\big(\rho_\eta^m\big), \tag{45}$$

*where $\sigma_\eta$ and $\rho_\eta = (1 - \eta + \eta\hat{\rho})^2$ are the constants of Theorem 4.6 and $K$ is defined in Theorem C.2.*

*Proof.* Set $\tilde{A} = \partial_x \varphi(\tilde{x}, \lambda)$ and $\tilde{c} = \partial_x f(\tilde{x}, \lambda)$. Let $\bar{\tilde{w}}_m$ be the deterministic KM sequence obtained by replacing every $\tilde{A}_m$ with $\tilde{A}$ and define the limit $\bar{\tilde{w}}_\infty = (I - \tilde{A})^{-1}\tilde{c}$.

Exactly as in the proof of Theorem C.2, decompose the error and bound the squared norm:

$$\mathbb{E}\big[\|\tilde{w}_m - \mathrm{d}_x f\|^2\big] \leq \mathbb{E}\big[\|\tilde{w}_m - \bar{\tilde{w}}_m\|^2\big] + \|(\bar{\tilde{w}}_m - \bar{\tilde{w}}_\infty) + (\bar{\tilde{w}}_\infty - \mathrm{d}_x f)\|^2$$
$$\leq \mathbb{E}\big[\|\tilde{w}_m - \bar{\tilde{w}}_m\|^2\big] + \|\bar{\tilde{w}}_\infty - \mathrm{d}_x f\|^2 + 2\|\bar{\tilde{w}}_m - \bar{\tilde{w}}_\infty\|\|\bar{\tilde{w}}_\infty - \mathrm{d}_x f\| + \|\bar{\tilde{w}}_m - \bar{\tilde{w}}_\infty\|^2.$$

With $A$ replaced by $\tilde{A}$ and $c$ by $\tilde{c}$, the variance bound in Theorem 4.6 yields $\mathbb{E}\|\tilde{w}_m - \bar{\tilde{w}}_m\|^2 \leq \sigma_\eta + O(\rho_\eta^m)$.

The resolvent identity Lemma C.1 gives $\bar{\tilde{w}}_\infty - \mathrm{d}_x f = (I - \tilde{A})^{-1}(\tilde{c} - c) + (I - \tilde{A})^{-1}(\tilde{A} - A)(I - A)^{-1}c$. Using $\|(I - \tilde{A})^{-1}\| \leq (1 - \rho)^{-1}$, $\|\tilde{A} - A\| \leq L_A \|\delta x\|$, and $\|\tilde{c} - c\| \leq L_c \|\delta x\|$ yields

$$\|\bar{\tilde{w}}_\infty - \mathrm{d}_x f\| \leq \Big(\frac{L_c}{1-\rho} + \frac{\|c\| L_A}{(1-\rho)^2}\Big)\|\delta x\|,$$

whose square is bounded by $K\|\delta x\|^2$.

Applying the deterministic part of Lemma A.5 to the single matrix $\tilde{A}$ gives

$$\|\bar{\tilde{w}}_m - \bar{\tilde{w}}_\infty\| = O(R^m).$$

Therefore, the cross product $2\|\bar{\tilde{w}}_m - \bar{\tilde{w}}_\infty\|\|\bar{\tilde{w}}_\infty - \mathrm{d}_x f\|$ because the first factor decays geometrically while the second is constant ($O(\|\delta x\|)$).

Collecting the three contributions gives the stated bound. $\qquad\square$

## C.3 Discussion

The two error bounds derived above share the *same structure* and, to leading order, the *same magnitude*. In both Theorems C.2 and C.3 the mean–square error can be written as

$$\underbrace{(\text{non-decaying error from variance})}_{\sigma_\eta \ \text{or} \ \frac{1+\alpha_\eta}{(1+\sqrt{\alpha_\eta})^2}\sigma_\eta} + \underbrace{K\|\delta x\|^2}_{\text{bias from inexact inner solution}} + \underbrace{(\text{decaying error})}_{O(\rho_\eta^m) \ \text{or} \ O(\max\{\rho_\eta,\alpha_\eta\}^m)}$$

where the constants $K$, $\sigma_\eta$, $\alpha_\eta$, and $\rho_\eta$ are identical (up to the harmless factor $(1+\alpha_\eta)/(1+\sqrt{\alpha_\eta})^2$) in the two approaches.

The key observation is that the variance component is unaffected by inner–loop bias. As shown in the proof of Theorem C.2, the perturbation $\delta x = \tilde{x} - x(\lambda)$ influences the estimator only through the resolvent difference, giving rise to the deterministic term $K\|\delta x\|^2$; it leaves unchanged the second-moment bound of the martingale difference sequence that drives the variance. Consequently, the variance-reduction mechanism built into MixedFP–KM is preserved even when the inner fixed-point equation is solved only approximately.

# D  Experiment Setup Details

In this section, we provide detailed information about the setup used in our empirical evaluations.

## D.1  Effect of Mixing Rate

The experiment in Section 5.1 evaluates the estimation error of hypergradients in the synthetic setting where the Jacobian matrix is a random variable. We implemented the following configuration:

The experiment used the following parameter settings: dimension of inner-parameter $d_x = 10$; dimension of outer-parameter $d_\lambda = 10$; number of parent matrices $H_1, \ldots, H_n$ for sampling $n = 1000$; eigenvalue bound parameter $\epsilon = 10^{-2}$; maximum iterations $M = 1000$; scale parameter values $\gamma \in \{10^{-3}, 10^{-2}, 10^{-1}, 1.0\}$; and mixing rate values $1 - \alpha \in \{0, 0.001, 0.01, 0.1, 1.0\}$, where $\alpha = 0$ corresponds to StocFP and $\alpha = 1.0$ corresponds to StocRB.

Any matrix in $H_1, \ldots, H_n$ was constructed to ensure that each eigenvalue follows a uniform distribution in $[0, 1 - \epsilon]$. The matrix $H_i \in \{H_1, \ldots, H_n\}$ was constructed through the following process: first, generating a random matrix $R \in \mathbb{R}^{d_x \times d_x}$ with entries sampled from a standard normal distribution; then creating a symmetric matrix $S = (R + R^\top)/2$; next, computing the QR decomposition of $S$ to obtain orthogonal eigenvectors $Q$; followed by sampling eigenvalues $\{\lambda_1, \ldots, \lambda_{d_x}\}$ uniformly from $[0, 1 - \epsilon]$; and finally constructing $H_i = Q \cdot \text{diag}(\lambda_1, \ldots, \lambda_{d_x}) \cdot Q^\top$.

The coefficients $c \in \mathbb{R}^{d_x}$, $d \in \mathbb{R}^{d_\lambda}$, and $B \in \mathbb{R}^{d_x \times d_\lambda}$ were sampled independently from a uniform distribution in $[0, 1]$ for each element.

## D.2  Compare with Existing Approaches

In this section, we provide detailed information about the setup and results in Section 5.2.

### D.2.1  Setups

The synthetic problem setting differs from that in Section 5.1 in the following aspects: we used fixed $\gamma = 1.0$ instead of varying it over $[0, 1]$, the dimensions of inner-parameter $x$ and outer-parameter $\lambda$ were both set to 100, we ran experiments with 10 different random seeds (instead of 100), and we used 10,000 iterations for each method. For the hyperparameter optimization task, we used binary classification with the Adult Income dataset. For the influence estimation task, we standardized the Fashion MNIST data with zero mean and unit variance. For the meta learning task, we used the min-max scaling on the California Housing dataset with the regularization parameter $\beta = 0.1$. Table 2 shows the dataset sizes and the values of $\gamma$ for $\varphi(x, \lambda) = (I - \gamma\hat{H})x + B\lambda$ used in the three real-world tasks.

| Task | Dataset | $n_{\text{train}}$ | $n_{\text{val}}$ | $d_x$ | $d_\lambda$ | $\gamma$ |
|---|---|---|---|---|---|---|
| Hyperparameter Optimization | Adult Income | 5000 | 5000 | 14 | 14 | $10^0$ |
| Influence Estimation | Fashion MNIST | 5000 | 5000 | 784 | 5000 | $10^{-2}$ |
| Meta Learning | California Housing | 5000 | 5000 | 8 | 8 | $10^{-1}$ |

Table 2: Experiment settings for the real-world tasks.

### D.2.2 Influence Estimation

We define the influence of the $j$-th instance $\Delta_j f$ as the change in validation loss $f(x, \lambda)$ after excluding the $j$-th instance from the training dataset with $N$ samples, denoted by

$$\Delta_j f = f(x(\mathbf{1}_{\neq j}), \mathbf{1}_{\neq j}) - f(x(\mathbf{1}), \mathbf{1})$$

where $\mathbf{1}_{\neq j} \in \{0, 1\}^N$ is a vector with ones except for a zero at the $j$-th index. Let $i \sim \text{Uniform}(\{1, \ldots, N\})$ denote an index of the instance in the dataset and $g(x, \lambda; i) = \lambda_i G(x; \xi_i)$ with $(\xi_i)_{i \leq N}$ represent the inner loss of the $i$-th instance $\xi_i$ masked by the $i$-th element of $\lambda$. Then, the inner solutions $x(\mathbf{1})$ and $x(\mathbf{1}_{\neq j})$ are defined as the minimizers of an with $\lambda = \mathbf{1}$ and $\lambda = \mathbf{1}_{\neq j}$, respectively:

$$x(\mathbf{1}) = \arg\min_x \mathbb{E}_i[g(x, \mathbf{1}; i)], \qquad x(\mathbf{1}_{\neq j}) = \arg\min_x \mathbb{E}_i[g(x, \mathbf{1}_{\neq j}; i)].$$

$x(\mathbf{1}_{\neq j})$ can be seen as the solution of the inner problems in which the $j$-th instance is excluded from the dataset, since $\mathbb{E}_i[g(x, \mathbf{1}_{\neq j}; i)] = \frac{1}{N}\sum_{i=1}^N g(x, \mathbf{1}_{\neq j}; i) = \frac{1}{N}\sum_{i=1}^N (1 - \delta_{ij}) G(x; \xi_i)$ where $\delta_{ij}$ is the Kronecker delta. Under this formulation, the $j$-th element of the hypergradient $-\mathrm{d}_{\lambda_j} f(x(\lambda), \lambda)$ at $\lambda = \mathbf{1}$ provides a linear approximation of the influence of the $j$-th instance, because

$$f(x(\mathbf{1}_{\neq j}), \mathbf{1}_{\neq j}) - f(x(\mathbf{1}), \mathbf{1}) = \mathrm{d}_\lambda f(x(\lambda), \lambda)\big|_{\lambda=\mathbf{1}}^\top (\mathbf{1}_{\neq j} - \mathbf{1}) + O(\|\mathbf{1}_{\neq j} - \mathbf{1}\|^2)$$

$$= -\mathrm{d}_{\lambda_j} f(x(\lambda), \lambda)\big|_{\lambda=\mathbf{1}} + O(1).$$

A similar definition of the influence and its approximation were also presented in Koh & Liang (2017). Recalling $\varphi(x, \lambda) = x - \frac{\gamma}{N}\sum_i \partial_x g(x, \lambda; i)$, at $\lambda = \mathbf{1}$, we have

$$\partial_x \varphi = I - \gamma H, \qquad \partial_{\lambda_j} \varphi = -\gamma \partial_x G(x; \xi_j),$$

where $H = \frac{1}{N}\sum_i \partial_x^2 G(x; \xi_i)$. From (2a), we have

$$-\mathrm{d}_{\lambda_j} f(x(\lambda), \lambda)\big|_{\lambda=\mathbf{1}} = -\partial_x G(x; \xi_j)^\top H^{-1} \partial_x f,$$

which is the downweight version of their Eq. (2).

For the initialization of inner-parameter $x$, we used samples from the normal distributions scaled by a constant factor for both logistic regression models used in the hyperparameter optimization and influence estimation tasks and linear regression models for the meta learning. For the outer-parameter $\lambda$, the initialization varies by task. In the synthetic task, $\lambda$ is initialized with values sampled uniformly from [0,1]. In the Hyperparameter optimization task, $\lambda$ serves as a regularization coefficient initialized with a small constant value. In the influence estimation task, $\lambda$ represents loss weights for each training sample, which is a ones-vector. In the meta learning task, $\lambda$ functions as a biased regularization term set to the initial value of the model's parameters $x$.

Any inner-problem optimization was performed using the Adam optimizer with a learning rate of 0.01. To rule out the effect incurred by inexact $x(\lambda)$, for any task, we used the full-batch inner loss to compute gradients for Adam and ran 1,000 epochs to ensure the convergence.

We optimized the hyperparameters using grid search to find the configuration that minimizes the average error over the last $M/10$ iterations. For each method, we performed a grid search over the following hyperparameter ranges:

- Mixing rate $1 - \alpha \in \{0, 10^{-4}, 2 \times 10^{-4}, 5 \times 10^{-4}, 10^{-3}, 2 \times 10^{-3}, 5 \times 10^{-3}, 10^{-2}, 2 \times 10^{-2}, 5 \times 10^{-2}, 10^{-1}, 2 \times 10^{-1}, 5 \times 10^{-1}\}$

- Stepsize $\eta \in \{10^{-3}, 2 \times 10^{-3}, 5 \times 10^{-3}, 10^{-2}, 2 \times 10^{-2}, 5 \times 10^{-2}, 10^{-1}, 2 \times 10^{-1}, 5 \times 10^{-1}, 1.0\}$

- For decreasing stepsize schedule: $\beta \in \{10, 20, 50, 100, 200, 500, 1000, 2000, 5000, 10000\}$ and $\delta \in \{10, 20, 50, 100, 200, 500, 1000, 2000, 5000, 10000\}$ with the constraint $\beta \leq \delta$

### D.3 Results

For completeness, we provide additional experimental results that complement the main findings presented in Section 5.2.

Fig. 3 presents the same experimental results as Fig. 2, but with error bars indicating the standard deviation across multiple runs. The error bars represent the standard deviation above and below the mean squared error across runs with different sequences $(\xi_m)_{m \in \mathbb{N}}$.

We also provide a computational cost comparison since our MixedFP and MixedFP-KM methods require computing two vector-Jacobian products (VJPs) per iteration,

Fig. 4 shows the results from Fig. 2 and Fig. 3 with the horizontal axis changed from algorithm iterations to the number of VJP computations. This transformation stretches the horizontal axis for MixedFP and MixedFP-KM by a factor of two, since our approach requires computing two VJPs per iteration: $\hat{A}_m \hat{u}_m$ and $\hat{A}_m \hat{w}_m$ as shown in (6b). This represents a worst-case analysis assuming VJP computation dominates the total cost and that our two VJPs require exactly twice the computation time.

The results show that our methods' advantage is reduced when evaluated on a wall-clock basis compared to iteration count, though performance remains competitive with existing approaches. Note that this represents a worst-case analysis assuming VJP computation dominates total cost and our two VJPs require exactly twice the computation time. In practice, the actual overhead is less than $2\times$ because the two VJP computations share common computations (particularly the forward pass, as they use the same samples). Moreover, the two VJP computations can be parallelized, potentially hiding the wall-clock impact.

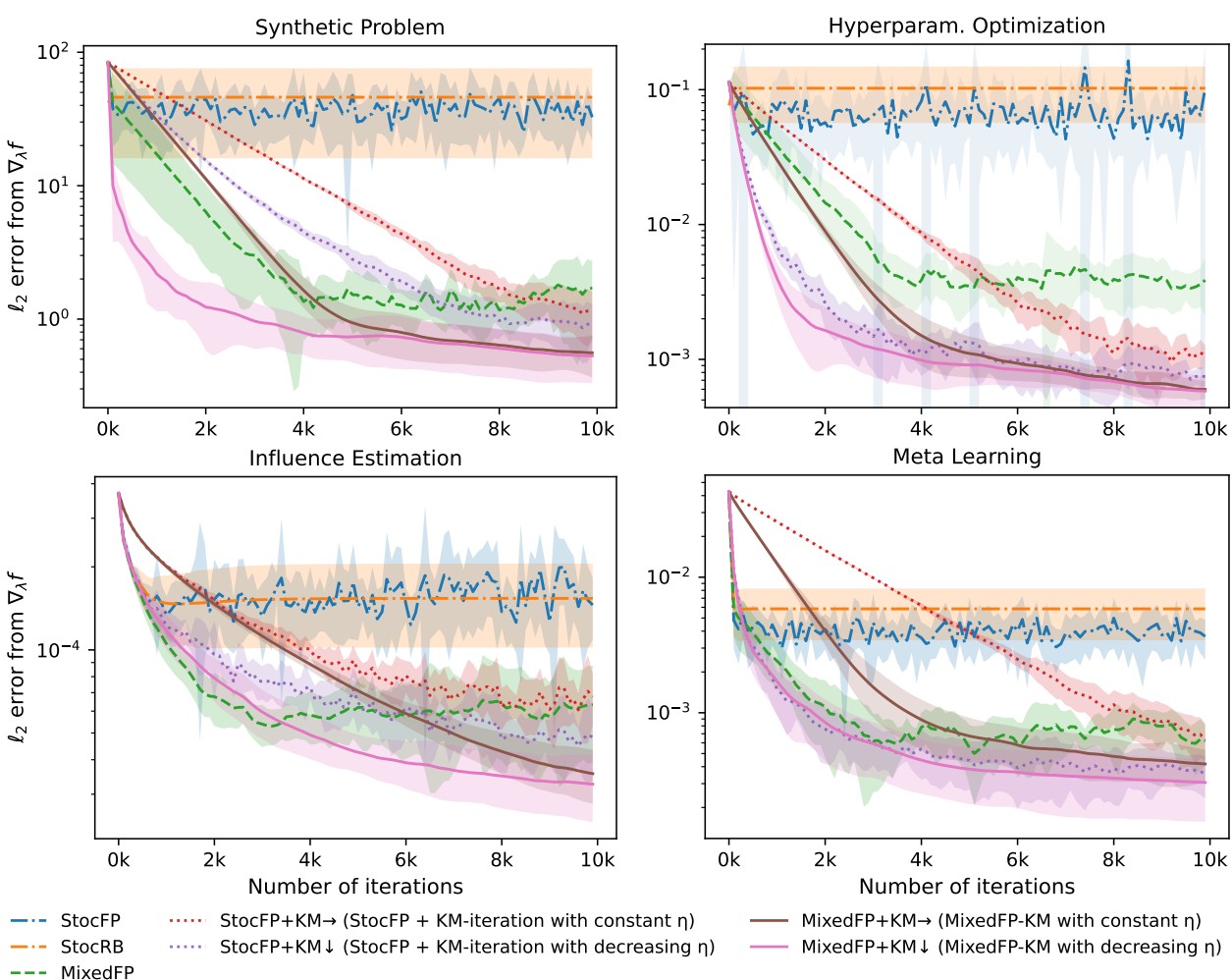

Figure 3: Results corresponding to Fig. 1 with added visualization of variance across runs. Each line shows the mean squared error over 10 trials with different Jacobian sample sequences $(\hat{A}_m)_{m \in \mathbb{N}}$, with error bars indicating standard deviation.

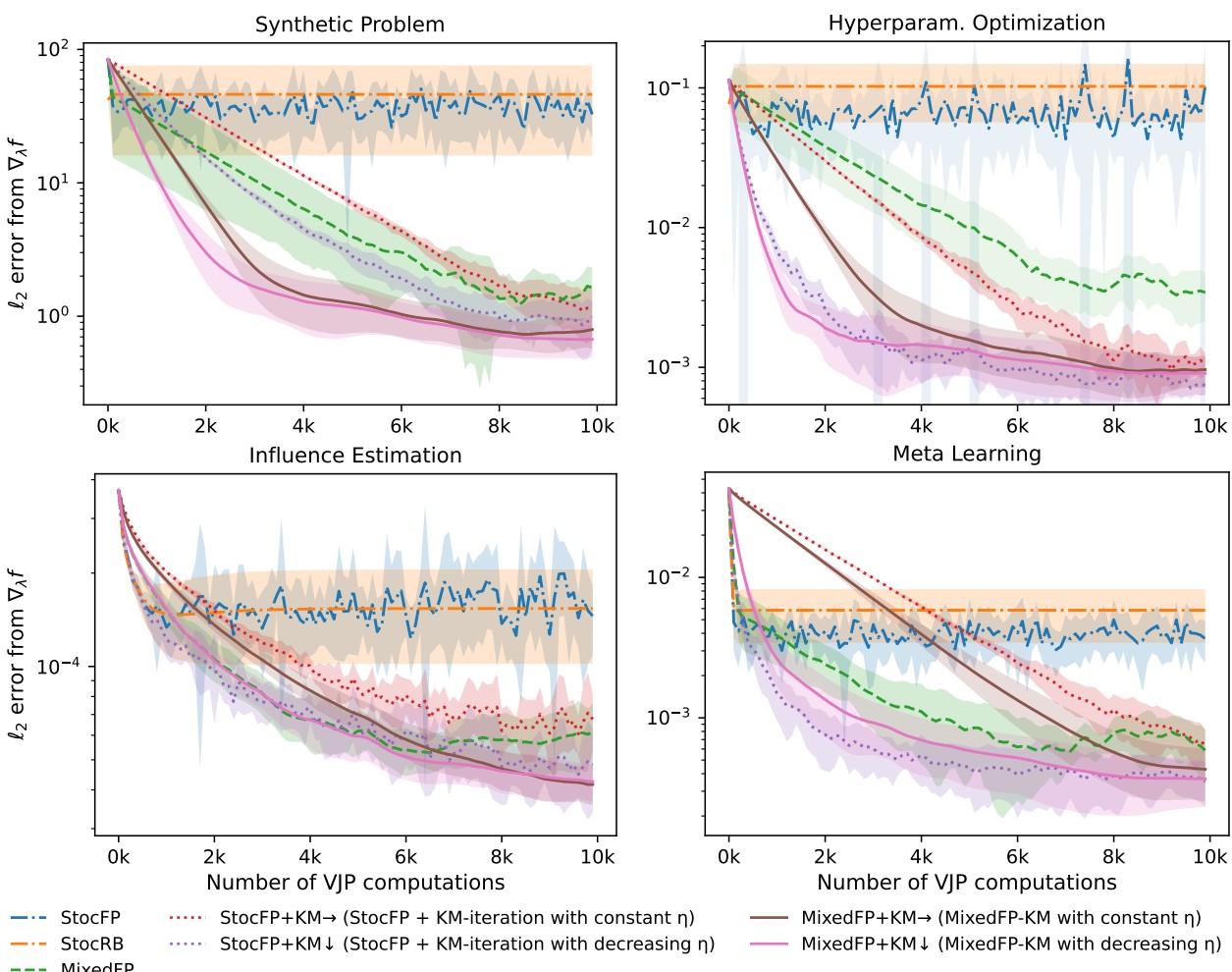

Figure 4: Comparison of stochastic hypergradient estimation methods evaluated by number of vector-Jacobian products (VJPs). The horizontal axis represents the number of VJPs, effectively stretching MixedFP and MixedFP-KM curves by a factor of two. This provides a worst-case analysis of wall-clock performance, assuming VJP computation dominates total cost and our two VJPs require exactly twice the computation time. Method-specific parameters were re-optimized under this new metric.

# E   Additional Experiments: Hyperparameter Optimization

In this section, we evaluate the practical impact of improved hypergradient estimation accuracy on actual bilevel optimization performance. While Section 5.2 demonstrated that our MixedFP-KM method achieves superior hypergradient estimation accuracy, it remains unclear whether this theoretical advantage translates to better optimization performance in real bilevel optimization tasks. To address this question, we conduct a bilevel optimization experiment using the hyperparameter optimization setting adopted in Section 5.2.

## E.1   Settings

We employ the same inner and outer problem formulation as the hyperparameter optimization task in Section 5.2, where the outer-parameter $\lambda$ represents regularization coefficients for each dimension of the inner-parameter $x$. The inner problem minimizes the binary cross-entropy loss on the training dataset, while the outer problem minimizes the validation loss with respect to the regularization parameters. In addition to the training and validation splits used in Section 5.2, we introduce a separate test set of 5,000 samples to evaluate the final model performance after the outer optimization.

We configure the bilevel optimization with 100 outer optimization steps using SGD with a learning rate of 20.0, and 100 inner optimization steps per outer iteration using Adam with a learning rate of 0.01. The hypergradient estimation depth is set to 100 iterations (reduced from 10,000 in Section 5.2 for computational efficiency), and we conduct experiments across five random seeds to get the averate test accuracy. We compare all hypergradient estimation methods from Section 5.2: `StocFP`, `StocRB`, `MixedFP`, `StocFP+KM→`, `StocFP+KM↓`, `MixedFP+KM→`, and `MixedFP+KM↓`, with all method-specific hyperparameters ($\eta$, $\alpha$, $\beta$, and $\delta$) set to the optimal values determined by grid search as in Section 5.2.

## E.2   Results and Discussion

Fig. 5 presents the learning curves for both validation and test losses across all methods. Methods incorporating the stochastic Krasnosel'skiĭ-Mann iteration (`MixedFP+KM↓`, `MixedFP+KM→`, and `StocFP+KM↓`) demonstrate substantially better performance than their non-KM counterparts (`StocRB`, `StocFP`, and `MixedFP`). Among these stochastic-KM-based methods, `MixedFP+KM↓` exhibits the steepest initial loss reduction and the highest test accuracy, which is correlated with the hypergradient estimation accuracy results shown in Fig. 6 where `MixedFP+KM↓` achieved the smallest error. These results validate that the improved hypergradient estimation accuracy demonstrated in Section 5.2 can translate to practical benefits in bilevel optimization.

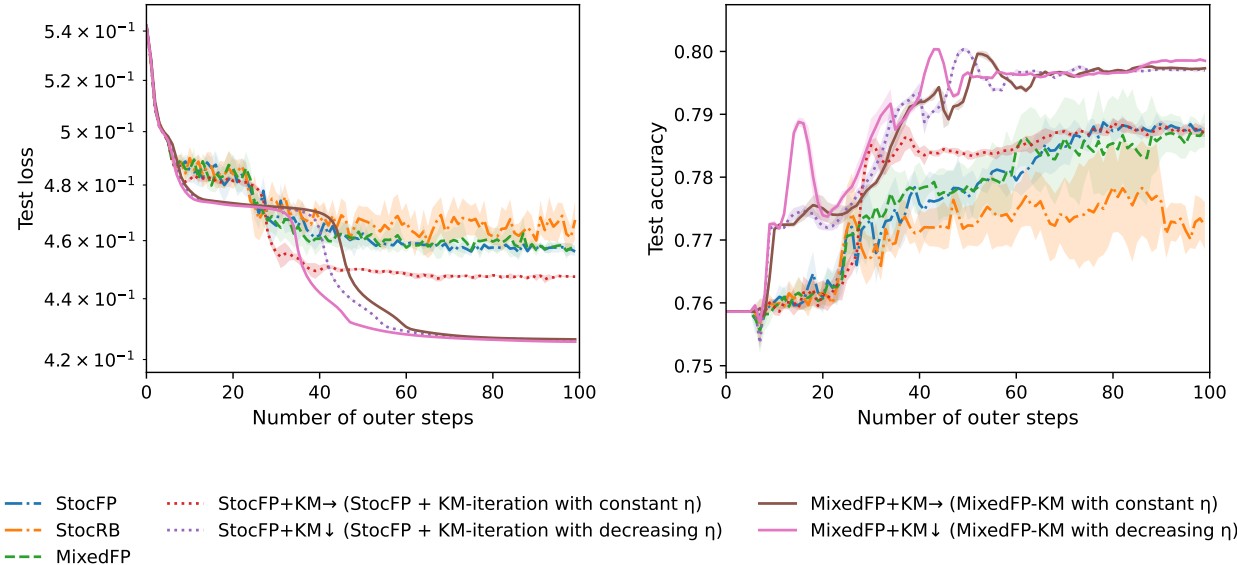

Figure 5: Test loss and accuracy curves for hyperparameter optimization of regularization coefficients across different hypergradient estimation methods. Curves show the average values over 5 runs with standard deviation error bars.

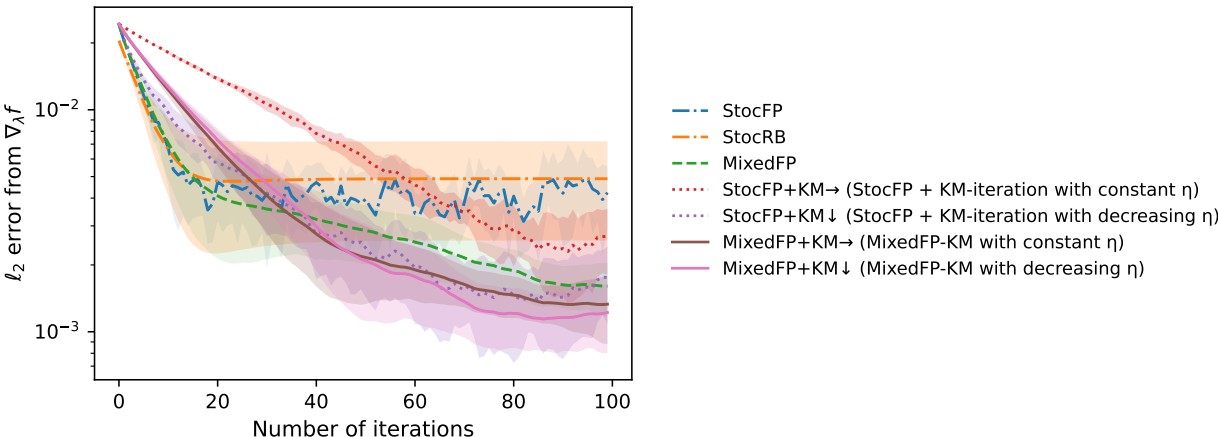

Figure 6: Hypergradient estimation errors at the initial outer-step using the optimal method-specific parameters ($\eta$, $\alpha$, $\beta$, and $\delta$) determined by grid search, which are then used in the bilevel optimization experiment shown in Fig. 5.

```python
def mixed_fp_km(
    f: Callable[[Tensor, Tensor], Tensor],
    phi: Callable[[Tensor, Tensor, Tensor], Tensor],
    x: Tensor,
    lambda_: Tensor,
    samples: Sequence[Tensor],
    etas: Sequence[float],
    alpha: float,
    M: int
) -> Tensor:
    # Compute outer gradients and initialize v, w, u
    v = torch.zeros_like(x)
    (c,) = torch.autograd.grad(f(x, lambda_), x)
    w, u = c.clone(), c.clone()

    for m in range(M):
        eta = etas[m]
        samp = samples[m]

        # phi with fixed lambda
        phi_fixed = lambda x_: phi(x_, lambda_, samp)

        # Compute VJPs
        _, (Aw,) = torch.autograd.functional.vjp(phi_fixed, (x,), (w,))
        _, (Au,) = torch.autograd.functional.vjp(phi_fixed, (x,), (u,))

        # Update v, w, u
        v = (1 - eta) * v + eta * (alpha * (v + u) + (1 - alpha) * w)
        w = (1 - eta) * w + eta * (Aw + c)
        u = (1 - eta) * u + eta * (Au)

    return v.detach()
```

Figure 7: Minimal Python implementation of the MixedFP-KM, corresponding to (9).

## F Python Implementation of MixedFP-KM

The following Python pseudocode Fig. 7 provides an explicit implementation of the MixedFP-KM algorithm, as introduced and discussed in Section 4.2.

In this implementation, the vector-Jacobian product for the stochastic mapping $\hat{A}_m = \partial_x \hat{\varphi}(x(\lambda), \lambda; \xi_m)$ is computed using the automatic differentiation tools of PyTorch. The iterative updates for the variables $(v, w, u)$ and other notations correspond to those given in (9).

