# OpenReview forum: "Variance Reduction of Stochastic Hypergradient Estimation by Mixed Fixed-Point Iteration"
_TMLR — Accepted by TMLR_

### Review · Reviewer_M2Gy · 2025-04-14

**Summary Of Contributions:**

The paper considers the problem of hypergradient estimation. It proposes MixedFP, a combination of the known algorithms StocFP and StocRB, which provably improves upon both algorithms with a faster convergence. Then, the authors show that the MixedFP algorithm can is actually the iterations of a contracting fixed point map. This enables them to improve MixedFP by proposing MixedFP-KM, which implements Krasnosel’skii-Mann iterations. Non-asymptotic $l^2$ error bounds are provided.
Numerical experiments of hypergradient estimation are provided on a synthetic problem, the hyperparameter optimization problem, the influence estimation problem, and meta-learning.

**Audience:**

Yes

**Claims And Evidence:**

Yes

**Requested Changes:**

### Clarity/precision
* **Confusion between outer cost $f$ and value function $\lambda\mapsto f(x(\lambda), \lambda)$**: The clarity could be improved by a clever choice of notation. Indeed, denoting $\nabla_\lambda f$ the hypergradient is confusing since it can be confused with the gradient of $f$ with respect to $\lambda$. Giving a name to the value function, that is, the function $\lambda \mapsto f(x(\lambda), \lambda)$ as classically done in the literature, would help to clarify the exposition. Moreover, page 3, the sentence "The hypergradient is the gradient of an outer-cost $f(x(\lambda), \lambda)\in\mathbb{R}$ with respect to the outer-parameter $\lambda\in\mathbb{R}^{d_\lambda}$, where the inner-parameter $x(\lambda)\in\mathbb{R}^{d_x}$" is really confusing, since the hypergradient is NOT the gradient of the outer function evaluated at $(x(\lambda), \lambda)$ but it is the gradient of *the composition of the outer function and the solution map of the inner fixed point problem*.

* **Implicit differentiation and approximate implicit differentiation**: Sometimes the paper calls "Approximate implicit differentiation" things that correspond to "Implicit differentiation" (without "approximate"). Implicit differentiation provides a closed-form formula of the hypergradient using the implicit function theorem. Approximate implicit differentiation is a set of techniques that compute an approximation of the exact hypergradient by replacing in the hypergradient formula, the exact value of the inner variable $x(\lambda)$ and the exact value of the linear system solution $(I-\partial_x \varphi(x(\lambda), \lambda))^{-1}\partial_x f(x(\lambda), \lambda)$ by approximation. For this reason, I think that the sentences "A method called approximate implicit differentiation (Pedregosa, 2016; Lorraine et al., 2019; Rajeswaran et al., 2019) considers the following form derived from the chain rule and implicit function theorem..." and "Approximate implicit differentiation (Lorraine et al., 2019; Pedregosa, 2016) uses the implicit function theorem and the chain rule to rewrite the hypergradient in a form that involves an inverse matrix..." should be rephrased.

* **Hessian-vector product**: Hessian-vector  (HVPs) are inappropriately mentioned on pages 4 and 6 for the following reasons:
    1. The paper considers the general setting where the inner problem is a fixed point iteration and not necessarily an optimization problem. As a consequence, the hypergradient computation in the different algorithms does not necessarily involve HVPs but involves Jacobian-vector products.
    2. Page 4:  "$\hat A_mw_m\in\mathbb{R}^{d_x}$ can be calculated in $O(d_x)$ time using the Hessian-vector product technique". The term "Hessian-vector product" only refers to the product between an Hessian matrix and a vector, without presuming the way it is computed. They can be computed naively by first computing the Hessian matrix and then multiplying it by the vector, or more efficiently using automatic differentiation techniques [1, 2]. Moreover, the $O(d_x)$ should be justified by a reference.

Thus, the sentences evoking HVPs should be rephrased in light of these two points.

* **Influence estimation experiment**: I don't get the setting of the influence estimation experiment. In particular, I don't see why the hypergradient catches the behavior of the performance when a sample is excluded. Moreover, I don't find the formulation of the problem proposed by the paper in the provided references [4, 5]. I think this part needs to be rewritten.

* **Page 3**: "which is equivalent to solving (2b) in Section 3.1 by stochastic gradient descent". It should be precised that, since $\varphi$ is contractive, solving (2b) is equivalent to minimizing a strongly convex quadratic function.

* **Page 3**: "$x(\lambda)$ is defined by the stationary point of a mapping..." -> "$x(\lambda)$ is defined as the fixed point of a mapping..."
* **Page 5**: "Section 4.2 explains that unbiased hypergradient estimation is achieved". I find this sentence misleading, since, due to a finite number of iterations, the $v_m$ is not an unbiased estimator of $\nabla_x f$. However, it is an asymptotically unbiased estimator.

### Experiments
* **Error bars**: the experiment results should report the variability of the results over the different trials.

* **Numerical comparison of the algorithms**: the complexity of an iteration of each algorithm is not identical. Therefore, comparing them with respect to wall-clock time rather than iteration number would be fairer.

* **Experiments with the full pipeline**: as mentioned in the weakness section, an experiment, for instance with hypergradient descent for hyperparameter selection, would enhance the paper.


### Appendix
I found some typos and some steps that should be a bit more detailed in the appendix, without affecting the final result.

* **Page 13, proof of lemma 4.3**: It seems that there is a confusion between $F$ and $T$.
* **Theorem 4.4**: If the $\sigma_2$ of the paper is the same as $\sigma_2$ in [3], then a square is missing on the denominator.
* **Appendix A3.**: Precise explicitly that the fixed step-size case is considered in this section to avoid confusion.
* **Lemma A.1**: Maybe I missed it but I did not find the definition of $\bar{\alpha}$ (which I presume is equal to $1-\alpha$).
* **Induction in Lemma A.1**: At the induction step, (19) is assumed for a given $m\geq 0$. Then it is shown that $v_{m+1} = \eta \sum_{k=0}^m r^{m-k}(\alpha u_k + \bar\alpha w_k)$ which is nothing else than (19) which is assumed to be true. To be rigorous, the induction step should show that, given $(19)$, we have $v_{m+2} = \eta \sum_{k=0}^{m+1} r^{m+1-k}(\alpha u_k + \bar\alpha w_k)$. This remark holds also for (20) and (21).
* **Page 18**: Why $\lVert\hat A_j - A\rVert^2\leq \hat\rho^2$?
* **Equation (38)**: Should $\lVert \hat A_j - A\rVert^2$ be removed in the right-hand-side?
* **Page 19**: It is not obvious to me why the equality holds $\mathbb{E}\left[\lVert \hat X_{j,0} + \bar\alpha \hat Y_j\rVert^2\right] = \left(\left(1-\bar\alpha \frac\eta{1-\hat R}\right)\hat R^j + \bar\alpha \frac\eta{1-\hat R}\right)^2$, thus it deserves a bit more details.
* **Page 19**: $\bar \alpha \frac\eta{1-R} = (1-\alpha) \frac1{1-\hat \rho}$: $R$ should be replaced by $\hat R$.
* **Pages 20 and 21**: Contrary to what is written, the inverse matrix identity is useless to get $\eta(I-P)^{-1}c = \nabla_x f$ since it is straightforward from the definition of $P$. But it is useful after when handling the term $P^kc + \bar\alpha\eta \sum_{s=0}^{k-1} P^{k-s-1}c$. Thus the sentence introducing the identity should be moved to the correct place.
* **Page 21, first equation**: $c$ is missing in the r.h.s. of the first line.


[1] Barak A Pearlmutter. Fast exact multiplication by the hessian. In *Neural computation*. 1994.

[2] A. Griewank and A. Walther. *Evaluating Derivatives: Principles and Techniques of Algorithmic Differentiation*, Second Edition. Society for Industrial and Applied Mathematics, 2008.

[3] R. Grazzi, M. Pontil, and S. Salzo. *Convergence properties of stochastic hypergradients*. In AISTATS, 2021

[4] P. Koh and P. Liang. *Understanding black-box predictions via influence functions*. In IICML, 2017.

[5] R. Khanna, B. Kim, J. Ghosh, and S. Koyejo. *Interpreting black box predictions using fisher kernels*. In AISTATS, pp. 2019.

**Strengths And Weaknesses:**

### Strengths
* The proposed hypergradient estimator is a simple modification of known estimator which achieves a more accurate and faster hypergradient approximation. This improvement is performed without any supplementary oracle computation.

* Numerical experiments corroborate the theoretical findings.

* The estimator comes with sound theoretical guarantees.

### Weaknesses
* **Clarity and precision**: Clarity has to be improved, in particular by clever notation choices (see requested changes for details) and better wording.

* **Approximation in $x(\lambda)$**: In the hypergradient approximation problem, there are two bottlenecks: the resolution of the linear system (which is treated in the paper) and the approximation of $x(\lambda)$. However, the approximation of $x(\lambda)$ influences the hypergradient precision, also in the linear system since the exact hypergradient uses $(I-\partial_x \varphi(\textcolor{red}{x(\lambda)}, \lambda))^{-1}\partial_x f(\textcolor{red}{x(\lambda)} \lambda)$ while the approximate hypergradient uses an approximation of $(I-\partial_x \varphi(\textcolor{red}{\tilde x}, \lambda))^{-1}\partial_x f(\textcolor{red}{\tilde x}, \lambda)$ where $\tilde x$ is an approximation of $x(\lambda)$. This approximation is not accounted for in the analysis while being fundamental in hypergradient estimation.

* **One-shot estimation**: Related to the previous weakness, the paper only studies one-shot estimation of the hypergradient. For instance, for the different experiments, comparing the performances of the whole pipeline when this hypergradient estimator is plugged would have been interesting (that is, not only the hypergradient error but also for example the performance in optimization for the hyperparameter optimization task).

* **Reproducibility**: The source code of the experiment is not provided, degrading the reproducibility.

---

> ### Author Response · Authors · 2025-06-27
> **Response to Reviewer M2Gy**
>
> We sincerely appreciate the thorough and constructive review.
> As mentioned in our official comment, we have substantially revised the manuscript to address the feedback.
> Below we provide detailed responses to each concern.
>
> ## 1. Approximation in the inner solution $x(\lambda)$
> > **(Reviewer comment)** *"The approximation of $x(\lambda)$ influences the hypergradient precision... This approximation is not accounted for in the analysis."*
>
> We appreciate this important observation.
> We have added **Appendix C** which provides a comprehensive theoretical analysis of how inexact inner solutions affect hypergradient estimation accuracy.
>
> Our analysis reveals that inner solution approximation errors will only affect the deterministic bias term (i.e., the convergence target) but do not impact the variance term that our method is specifically designed to reduce. Furthermore, the effects of inner solution approximation are identical across existing methods (StocFP, StocRB) and our proposed MixedFP-KM, preserving our method's advantages.
>
> While this means our approach does not provide additional benefits for handling inner solution approximation errors, this analysis clarifies the specific characteristics and limitations of our method.
> We thank the reviewer for bringing this important aspect to our attention.
>
>
> ## 2. One-shot estimation vs full pipeline evaluation
>
> Our primary focus is variance reduction in hypergradient estimation itself, which is crucial for tasks like influence estimation where the hypergradient itself is of interest.
> However, we acknowledge the importance of end-to-end evaluation for a full pipeline such as the bilevel optimization.
>
> **Appendix E (Additional Experiments: Hyperparameter Optimization)** is added to our revised manuscript, which includes experiments that evaluate our method in a complete hyperparameter optimization pipeline, demonstrating that our variance reduction can translate to improved convergence in bilevel optimization.
>
> ## 3. Clarity and Precision Issues
>
> ### a. *"Confusion between outer cost $f$ and value function $\lambda \mapsto f(x(\lambda),\lambda)$"*
>
> We have clarified the notation in Section 3.1 by providing explicit definitions for the value function and establishing clear distinctions between the partial derivative $\partial\_{\lambda}f(x(\lambda),\lambda)$ and the hypergradient $\mathrm{d}\_{\lambda}f(x(\lambda),\lambda)$.
> The revised text in Section 3.1 now states:
>
> > **(Section 3.1)** *"The hypergradient $\mathrm{d}\_{\lambda} f(x(\lambda),\lambda)$ is **the total derivative of the composite function** $\lambda \in \mathbb{R}^{d\_\lambda} \mapsto f(x(\lambda),\lambda)\in \mathbb{R}$, ..."*
>
> > **(Section 3.1)** *"... $\partial\_{\lambda}f$ represents **the partial derivative of $f(x(\lambda),\lambda)$ with respect to the second argument, treating $x(\lambda)$ as a constant.** "*
>
> ### b. *"Implicit differentiation and approximate implicit differentiation"*
>
> > **(Reviewer comment)** *"Sometimes the paper calls 'Approximate implicit differentiation' things that correspond to 'Implicit differentiation' (without 'approximate')."*
>
> We have clarified this distinction throughout the manuscript, for example in Section 3.1. The revised text now states:
>
> > **(Section 3.1)** *"Approximate implicit differentiation (AID) (Lorraine et al., 2019; Pedregosa, 2016) is a **set of approaches** consisting of the **derivation of a linear system for $\mathrm{d}\_{\lambda} f(x(\lambda),\lambda)$ using the implicit function theorem** along with two approximations: an **approximate inner solution** $x(\lambda)$ and an **approximate linear system solution**."*
>
>
> ### c. *"Hessian-vector product: Hessian-vector (HVPs) are inappropriately mentioned ..."*
>
> We have corrected this throughout the manuscript.
> In Section 3.2.1, we now state:
>
> > **(Section 3.2.1)** *"A key advantage of this method is that the **vector-Jacobian product** $\hat{A}\_{m} w\_{m} \in \mathbb{R}^{d\_x}$ can be calculated in $O(d\_x)$ time using **reverse mode automatic differentiation (Baydin et al., 2018)**, because $\hat{A}\_{m} w\_{m} $ can be seen as the derivative of the scalar function $x \mapsto \varphi(x,\lambda;\xi\_m)^{\top} w\_m$ evaluated at $x(\lambda)$."*
>
> We chose the term "**vector-Jacobian product**" (or "transposed Jacobian-vector product"), given the definition of our Jacobian which is transposed from the common notation of the Jacobian matrix (Section 3.1).
> This terminology accurately distinguishes it from the **Jacobian-vector product**, which can be efficiently computed using ***forward mode*** automatic differentiation (Baydin et al., 2018).

---

> > ### Author Response · Authors · 2025-06-27
> >
> > ### d. Influence Estimation &mdash; *"I don't see why the hypergradient catches the behavior of the performance."*
> >
> > We have included details on the connection between influence estimation and the hypergradient estimation in our experimental section (**Section 5.2.1**).
> > We also added **Appendix E.2**, which provides more comprehensive background and clarifies the correspondence to the existing work from Koh & Liang (2017).
> >
> > ### e. Correction in Related Work
> > > **(Reviewer comment)** *""which is equivalent to solving (2b) in Section 3.1 by stochastic gradient descent". It should be precised that, since $\varphi$ is contractive, solving (2b) is equivalent to minimizing a strongly convex quadratic function.."*
> >
> > We have corrected this in the related work section:
> > > **(Section 2)** *"which is equivalent to solving $ \min\_v \frac{1}{2} \| (I- \partial\_{x} \varphi)v - \partial\_{x} f\|^2$ by the stochastic gradient descent (Arbel & Mairal, 2022)."*
> >
> > ### f. Misleading Use of "Unbiased Estimation"
> > > **(Reviewer comment)** *"Section 4.2 explains that unbiased hypergradient estimation is achieved". I find this sentence misleading, since, due to a finite number of iterations, the $v\_m$ is not an unbiased estimator of $\nabla\_x f$. However, it is an asymptotically unbiased estimator."*
> >
> > We thank the reviewer for this important clarification.
> > We have revised the terminology throughout the manuscript to use the more accurate expression "**almost sure convergence to the true hypergradient**" instead of "unbiased hypergradient estimation".
> >
> > > **(Section 4.2)** *"This section shows that the stochastic KM iteration can be applied to MixedFP, enabling almost sure convergence to the true hypergradient."*
> >
> > While "asymptotically unbiased estimator" would also be technically correct, it also applies to existing methods (StocFP and StocRB).
> >
> >
> > ## 5. Experiments
> >
> > ### a. Error bars in figure
> >
> > We have included error bars in the results shown in Figure 3 of Appendix D.3.
> >
> > ### b. Numerical comparison of the algorithms
> >
> > > **(Reviewer comment)** *"Numerical comparison of the algorithms: the complexity of an iteration of each algorithm is not identical. Therefore, comparing them with respect to **wall-clock time** rather than iteration number would be fairer.*"
> >
> > We have addressed this concern by providing computational cost analysis in **Appendix D.3**. Since MixedFP requires two vector-Jacobian products per iteration (compared to one for StocFP and StocRB), comparing methods by iteration count does not fairly reflect computational cost.
> >
> > In Appendix D.3, we present a "worst-case analysis" for our method where we scale the x-axis by a factor of 2 to account for the doubled VJP computation cost per iteration. Even under this conservative assumption that completely ignores potential parallelization benefits and assumes identical VJP costs, our method remains competitive.
> >
> > Additionally, we note that the results in Fig. 1 should represent a fair comparison from the perspective of sampling cost.
> > Since all methods use the same number of samples per iteration, Fig. 1 effectively shows the difference in estimation error under equal sampling budgets.
> >
> > ## 6. Appendix Corrections
> >
> > We appreciate the reviewer's careful attention to technical details. All identified issues have been corrected in the revised manuscript, which significantly improves the paper's accuracy and rigor. We provide detailed explanations for the two most critical points below:
> >
> > ### a. Bound $\lVert\hat A\_j - A\rVert^2\leq \hat\rho^2$
> >
> > The reviewer correctly identified that the bound $\|\hat{A}\_j - A\|^2 \leq \hat{\rho}^2$ is incorrect for spectral norm, as this bound only holds for a norm induced by the inner product.
> >
> > We have corrected this by directly bounding the full vector norm $\|(\hat{A}\_j - A)(\hat{X}\_{j,0} + \hat{Y}\_j)c\|^2$ without decomposing it, which maintains the same conclusion. This correction appears just before equation (37) in the proof of Lemma A.4. We have also applied the same correction to the existing method proof (Theorem 4.6).
> >
> > ### b. Equality $\mathbb{E}\left[\lVert \hat X\_{j,0} + \bar\alpha \hat Y\_j\rVert^2\right] = \left(\left(1-\bar\alpha \frac\eta{1-\hat R}\right)\hat R^j + \bar\alpha \frac\eta{1-\hat R}\right)^2$
> >
> > The reviewer correctly identified that this should be an inequality rather than an equality.
> > We have also corrected the subsequent analysis considering the cross-term as follows:
> > $$
> > \mathbb{E}\left[\| \hat{X}\_{j,0} +\overline{\alpha }\hat{Y}\_{j} \| ^{2} \right] \le\left(\left( 1-\overline{\alpha }\frac{\eta }{1-\hat{R}}\right)\hat{R}^{j} +\overline{\alpha }\frac{\eta }{1-\hat{R}}\right)^{2}
> >  \leq 2\left( 1-\overline{\alpha }\frac{\eta }{1-\hat{R}}\right)^{2} \hat{R}^{2j} +2\left(\overline{\alpha }\frac{\eta }{1-\hat{R}}\right)^{2} ~~\text{(Page 20)}
> > $$
> > which slightly affects the results for both our method and the existing methods (Theorem 4.1 and 4.6).

---

> > > ### Author Response · Authors · 2025-06-27
> > >
> > > ## 7. Reproducibility
> > >
> > > We have uploaded our source code as supplementary material.
> > > Code will be made public on github once accepted.
> > >
> > > ---
> > >
> > > **Reference**
> > >
> > > [Koh & Liang (2017)] P. Koh and P. Liang. Understanding black-box predictions via influence functions. In ICML, 2017.
> > >
> > > [Baydin et al. (2018)] Baydin, Atilim Gunes, et al. "Automatic differentiation in machine learning: a survey." JMLR 18.153 (2018): 1-43.

---

### Review · Reviewer_8hMc · 2025-04-28

**Summary Of Contributions:**

The paper proposes MixedFP, a stochastic hypergradient estimation method that combines StocFP and StocRB updates to reduce the variance in hypergradient estimation. The authors further apply the stochastic Krasnosel’skiĭ-Mann iteration to MixedFP to guarantee almost sure convergence to the true hypergradient. Theoretical analyses provide variance bounds, and empirical results on synthetic and real-world tasks support the approach.

**Audience:**

Yes

**Broader Impact Concerns:**

Not applicable.

**Claims And Evidence:**

Yes

**Requested Changes:**

- Clearly acknowledge that MixedFP requires updating multiple iterates per step, resulting in increased per-iteration computational cost compared to StocFP and StocRB.
- Explicitly state that the theoretical analysis for decreasing stepsizes is limited to asymptotic convergence, without non-asymptotic error bounds, and reflect this limitation earlier in the main text.

Also, some suggestions :
- Report experimental comparisons based on wall-clock time or normalized computational effort, rather than iteration count alone, to ensure a fair evaluation.
- Discuss that the experimental setting assumes full-batch convergence of the inner problem, which does not reflect the more challenging stochastic or approximate scenarios where hypergradient estimation is typically critical.
- Briefly comment on the practical difficulty of satisfying the unbiased Jacobian estimation assumption in real-world large-scale applications.

**Strengths And Weaknesses:**

**Strengths**:
- The idea to construct a mixed estimator is natural and carefully executed.
- The mathematical derivations are sound and based on well-established tools from stochastic approximation and fixed-point theory.
- The theoretical analysis precisely quantifies the error bounds and clearly identifies when the proposed method offers improvements.

**Weaknesses**:
- The improvement in variance bounds is modest.
- While MixedFP shares the same order of computational complexity as StocFP and StocRB, it requires updating and maintaining multiple iterates at each step, resulting in increased per-iteration computational cost.
- The theoretical analysis for decreasing stepsizes is limited to asymptotic convergence, without non-asymptotic error bounds, while decreasing stepsizes are used in some experiments.

---

> ### Author Response · Authors · 2025-06-27
> **Response to Reviewer 8hMc**
>
> Thank you for your thorough review.
> We appreciate your positive feedback on our mathematical analysis and have revised the manuscript to address your concerns.
> Here are our responses:
>
> ---
> ## 1. Computational Cost and Per-Iteration Overhead
>
> ### a. Acknowledgment of Computational Cost
>
> > **(Reviewer comment)** *"Clearly acknowledge that MixedFP requires updating multiple iterates per step, resulting in increased per-iteration computational cost compared to StocFP and StocRB."*
>
> You are absolutely correct, and we have explicitly acknowledged this limitation.
> As you noted, MixedFP requires two vector-Jacobian products (VJPs) per iteration compared to one for StocFP and StocRB. We now explicitly state this in the main text:
>
> > **(Section 4.1)** *"... it requires two vector-Jacobian-products at each step. **This results in a higher per-iteration computational cost compared to StocFP and StocRB**, which only requires a single JVP computation."*
>
> ### b. Experiments for Computational Cost
>
> > **(Reviewer comment)** *"Report experimental comparisons based on wall-clock time or normalized computational effort, rather than iteration count alone, to ensure a fair evaluation."*
>
> **We have addressed this concern by providing computational cost analysis in Appendix D.3.**
> Since MixedFP requires two vector-Jacobian products per iteration (compared to one for StocFP and StocRB), comparing methods by iteration count does not fairly reflect computational cost.
>
> In Appendix D.3, we present a "worst-case analysis" for our method where we scale the x-axis by a factor of 2 to account for the doubled VJP computation cost per iteration.
> Even under this conservative assumption that completely ignores potential parallelization benefits and assumes identical VJP costs, our method remains competitive.
>
> Additionally, we note that the results in Fig. 1 should represent a fair comparison from the perspective of sampling cost.
> Since all methods use the same number of samples per iteration, Fig. 1 effectively shows the difference in estimation error under equal sampling budgets.
>
> ## 2. Theoretical Analysis Limitations for Decreasing Step Sizes
>
> > **(Reviewer comment)** *"The theoretical analysis for decreasing stepsizes is limited to asymptotic convergence, without non-asymptotic error bounds, while decreasing stepsizes are used in some experiments."*
>
> We now explicitly state that:
>
> > **(Section 4.2)** *"While Theorem 4.5 provides non-asymptotic error bounds for fixed step sizes, our analysis for decreasing step sizes (Theorem 4.4) is limited to asymptotic convergence guarantees without finite-time bounds."*
>
> ## 3. Full-Batch Inner Problem Assumption
>
> > **(Reviewer comment)** *"The experimental setting assumes full-batch convergence of the inner problem, which does not reflect the more challenging stochastic or approximate scenarios where hypergradient estimation is typically critical."*
>
> We have added **Appendix C** which provides theoretical analysis of how inexact inner solutions affect hypergradient estimation accuracy.
> Due to time constraints, we were unable to experimentally demonstrate the relationship between inner-problem convergence and hypergradient estimation error.
> We have explicitly added clarification in our experimental settings that this aspect remains unvalidated experimentally:
>
> > **(Section 5.2.2)** *"We approximated $x(\lambda)$ as accurately as possible by minimizing $g$ using full-batch optimizations until they converge. The impact of approximation errors in $x(\lambda)$ is analyzed only theoretically in Appendix C."*
>
>
> ## 4. Unbiased Jacobian Estimation Assumption
>
> > **(Reviewer comment)** *"Briefly comment on the practical difficulty of satisfying the unbiased Jacobian estimation assumption in real-world large-scale applications."*
>
> While we cannot provide specific examples of when this assumption becomes problematic, we have added discussion on the validity of this assumption in practice:
>
> > **(Section 3.2)** *"Assumption 2(i) can be easily satisfied in a typical machine learning scenario where the stochastic mapping is a stochastic gradient descent, i.e., $\hat{\varphi}(x,\lambda;\xi) = x - \gamma \partial_x \hat{g}(x,\lambda;\xi)$, and the instance $\xi$ is uniformly sampled from a finite dataset."*
>
> ---
>
> We appreciate your thorough feedback, which has significantly improved the manuscript's clarity and practical relevance while maintaining scientific rigor.

---

### Review · Reviewer_StT7 · 2025-06-09

**Summary Of Contributions:**

The paper presents a new estimation technique of hypergradients from stochastic estimates that achieves variance reduction over previous techniques by mixing them. The authors detail clearly two previous methods and how they propose to combine them. In addition, they propose to blend their method in Krasnosel’ski˘ı-Mann iterations to further enhance it. They provide theoretical bounds on the expected mean squared error showing improvements over previous methods. They provide numerical experiments showcasing the advantages of the method that their theory forecasted.

**Audience:**

Yes

**Claims And Evidence:**

Yes

**Requested Changes:**

Question:
- "The error comprises two distinct decay rates, and for sufficiently large m, the slower decay rate eventually dominates." I don't understand: for a given problem $\rho_1$ and $\alpha$ are fixed so there is no reason for the decay rate to switch from one decay to the other.
- "While both StocFP (4) and StocRB (5) provide the same hypergradient approximation in expectation, they yield different biases". Biases are generally the difference between the expected estimator and the true estimate. So if they are the same in expectation, I would expect them to have same biases. Did you want to speak about e.g. their variance? Sorry if I'm missing something obvious.

Typos:
- $\sigma_1$ is used both Theorem 4.1 and Theorem 4.2, consider using two different notations (same for $\sigma_2$
- Maybe use $\eta_m$ instead of $\eta_k$ in the sums of Theorem 4.2
- In Lemma 4.3, say "the function $F(z)=...$", avoid using math symbols as entities of a phrase.

**Strengths And Weaknesses:**

**Strengths**
- The paper is extremely well written. From the choice of the notations to the comparison side by side of the algorithms, the authors managed to provide a very clear manuscript to an idea that is actually quite technical.
- The application may be slightly niche, but the idea is original and the results demonstrate clearly the advantage of their method on several benchmarks.
- The theory gives interesting insights that are well illustrated with carefully crafted experiments.

**Weaknesses**
- A snippet of code could be beneficial for the paper to grab adoption. As the authors mention, the implementation requires typically hvps (or vjps of the implicit equation) that readers could be unfamiliar with.

---

> ### Author Response · Authors · 2025-06-27
> **Response to Reviewer StT7**
>
> We sincerely thank the reviewer for the positive evaluation and constructive feedback. We appreciate your recognition of our paper's clarity and originality. As mentioned in our official comment, we have substantially revised the manuscript to address the feedback. Below we provide detailed responses to each concern.
>
> ---
> ## 1. Code Availability for Adoption
>
> > **(Reviewer comment)** *"A snippet of code could be beneficial for the paper to grab adoption. As the authors mention, the implementation requires typically hvps (or vjps of the implicit equation) that readers could be unfamiliar with."*
>
> We have fully addressed this concern in three ways.
> - **Enhanced VJP explanation in main text**: The main text now includes a clearer explanation of vector-Jacobian products to aid understanding. As noted in Section 3.2.1:
>     > **(Section 3.2.1)** *"the vector-Jacobian product $\hat{A}\_{m} w\_{m} \in \mathbb{R}^{d\_x}$ can be calculated in $O(d\_x)$ time using reverse mode automatic differentiation (Baydin et al., 2018), because $\hat{A}\_{m} w\_{m} $ is the derivative of the scalar function $x \mapsto \varphi(x,\lambda;\xi\_m)^{\top} w\_m$ evaluated at $x(\lambda)$."*
> - **Complete source code upload**: Our complete source code has been uploaded as supplementary material to reproduce all experimental results.
> - **Minimal implementation in paper**: **Appendix F** has been added with a minimal Python implementation of MixedFP-KM using PyTorch to facilitate adoption.
>
> ## 2. Questions and Clarifications
>
> ### a. Decay Rate Switching
>
> > **(Reviewer comment)** *"The error comprises two distinct decay rates, and for sufficiently large m, the slower decay rate eventually dominates." I don't understand: for a given problem $\rho\_1$ and $\alpha$ are fixed so there is no reason for the decay rate to switch from one decay to the other.*
>
> You are correct to point out this confusion in our explanation.
> Our original explanation in the main text was not sufficient, requiring readers to examine the appendix proof to understand this behavior. The bound $O(\max\\{\rho\_1,\alpha\_1\\}^m)$ actually comes from the sum of two separate terms with decay rates $O(\rho\_1)$ and $O(\alpha\_1)$ (as shown in equation (39) of Lemma A.4).
> Initially both terms contribute, but eventually the slower-decaying term dominates.
>
> We have clarified this in the revised manuscript by adding the following explanation
>
> > **(Section 5.1.2)** *"This seems to reflect the bound $O(\max\\{\rho\_1,\alpha\_1\\}^m)$, which is yielded from the sum of two terms with different decay rates $\rho\_1$ and $\alpha\_1$ (as in Appendix A.5 (39)); the term with the faster decay rate converges quickly, and for sufficiently large $m$, the slower decay rate eventually dominates."*
>
> ### b. "Different Biases" Terminology
>
> > **(Reviewer comment)** *"While both StocFP (4) and StocRB (5) provide the same hypergradient approximation in expectation, they yield different biases". Biases are generally the difference between the expected estimator and the true estimate. So if they are the same in expectation, I would expect them to have same biases.*
>
> You are absolutely correct about the technical definition of bias.
> StocFP (4) and StocRB (5) actually have the same bias and variance.
> We have corrected this imprecise terminology in Section 3.2.2:
>
> > **(Section 3.2.2)** *"While both StocFP (4) and StocRB (5) provide the same hypergradient approximation in expectation, they yield **different estimates** given the same samples."*
>
> ## 3. Notation and Technical Corrections
>
> Thank you for the careful attention to these technical details.
> We have addressed all the notation and technical issues you identified.

---

### Decision · Action_Editor_Ke4x · 2025-07-23

**Recommendation:** Accept as is

**Audience:**

Yes

**Audience Explanation:**

Hypergradient estimation under stochastic inner loops is central to modern hyperparameter tuning, meta-learning, topics that frequently appear in TMLR and other related ML venues. Reviewer StT7 called the work "very well written" and M2Gy said it "may be of interest to some of the TMLR's audience".

**Claims And Evidence:**

Yes

**Claims Explanation:**

All three reviewers judged the theoretical and empirical backing to be correct. While some points of notation and minor proof details required correction, the authors’ rebuttal fixed those issues.